# CGES: Confidence-Guided Early Stopping for Efficient and Accurate Self-Consistency

## Abstract

Large language models (LLMs) are often queried multiple times at test time, with predictions aggregated by majority vote. While effective, this *self-consistency* (Wang et al., 2023) strategy requires a fixed number of calls and fails when the correct answer is infrequent. We introduce ***Confidence-Guided Early Stopping (CGES)***, a Bayesian framework that forms posteriors over candidate answers from scalar confidence signals—derived from token probabilities or reward models—and adaptively halts sampling once posterior mass exceeds a threshold. We provide theoretical guarantees in both the ideal case of perfectly calibrated confidences and the realistic regime with noisy confidences. Averaged over five reasoning benchmarks, CGES reduces the average number of calls by **69.4%** (e.g., from 16.0 to 4.9) while maintaining accuracy within **0.06 percentage points** of self-consistency.

## 1 Introduction

Large language models (LLMs) have achieved strong progress across reasoning, problem solving, and open-domain tasks. A common way to improve reliability is *test-time scaling* (Snell et al., 2025), a family of methods that allocate additional inference-time computation to improve performance. One subset of these methods samples multiple responses and aggregates them into a final prediction. Among the most widely used methods, self-consistency (SC) (Wang et al., 2023) aggregates outputs by majority vote, leveraging the intuition that the most frequent answer across diverse generations is likely to be correct. While simple and effective in many settings, majority-based aggregation suffers from two major shortcomings. First, it assumes that response frequency is a faithful proxy for correctness, which fails in cases where the correct answer appears infrequently. Second, it requires a fixed number of model calls regardless of confidence, leading to substantial inefficiency.

Confidence signals offer an alternative perspective. Instead of depending solely on frequency, one can incorporate confidence scores that capture the model's belief in each response. These scores may be derived from different sources. *Token probabilities* are taken directly from the model's output distribution and reflect how certain the model is about generating each token. *Calibration schemes* adjust these raw probabilities so that they better match the actual likelihood of correctness, turning overconfident or underconfident estimates into more reliable signals. *External reward models* are trained separately, often with human feedback or domain-specific supervision, and provide an independent measure of response quality beyond the model's own probabilities. *Such signals can distinguish between frequent but uncertain answers and rare yet confident ones*.

We propose a *confidence-based Bayesian framework* for test-time scaling. Our framework builds on the idea that incorporating confidence enables robust aggregation and adaptive stopping, where sampling stops once sufficient certainty is reached, reducing cost without sacrificing accuracy. Our approach computes posterior probabilities over candidate answers by treating each response and its associated confidence as probabilistic evidence. This yields two key advantages over majority voting: (1) when the correct answer is frequent, our method reaches the same conclusion but often stops earlier by exploiting high-confidence signals, improving efficiency; (2) when the correct answer is a minority, our method can still recover it by amplifying the influence of confident predictions, where majority voting fails. Figure 1 illustrates both phenomena on real items: our framework (a) halts after only a few calls when confidence concentrates, and (b) outperforms majority vote when the correct answer is a minority but highly confident. We further formalize this intuition by proving theoretical guarantees under ideal conditions where confidence scores are faithful to the true likelihood

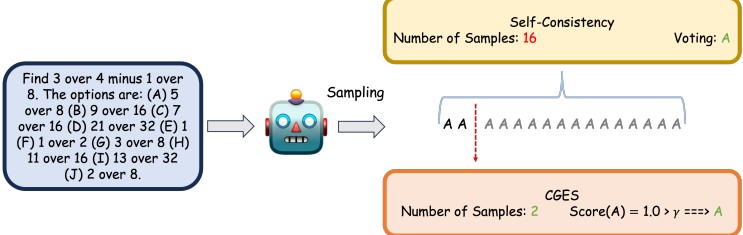

*(a) Efficiency: CGES stops early once its posterior exceeds $\gamma$, while SC uses a fixed budget ($B$=16).*

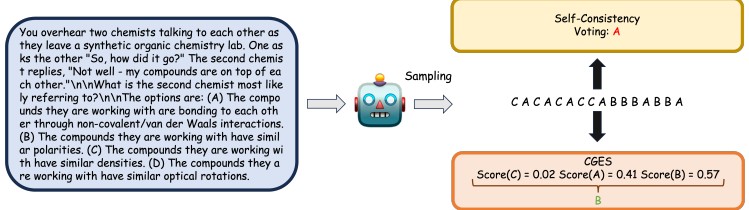

*(b) Accuracy: SC's majority vote is wrong, but CGES aggregates confidences and selects the correct answer.*

Figure 1: **Examples of CGES vs. SC.** Top: early stopping with high confidence; Bottom: recovering a minority-but-confident answer.

of correctness. We then extend the analysis to the more practical case of noisy confidence estimates, where scores may be imperfect reflections of true correctness.

To operationalize this framework, we introduce *Confidence-Guided Early Stopping (CGES)*, which integrates Bayesian scoring with an adaptive stopping rule. CGES enables accuracy–efficiency trade-offs by halting once posterior concentration exceeds a threshold or a budget is reached. We evaluate CGES on AIME24, MATH500 (Hendrycks et al., 2021), GSM8K (Cobbe et al., 2021), GPQA (Rein et al., 2024), and MMLU_Pro (Wang et al., 2024), comparing against self-consistency (Wang et al., 2023) and early-stopping self-consistency (Li et al., 2024). CGES consistently reduces LLM calls by large margins while maintaining or improving accuracy, demonstrating the value of calibrated confidence for test-time scaling and principled Bayesian aggregation. Our contributions are as follows:

- We propose a Bayesian framework that incorporates confidence estimates into self-consistency, enabling more accurate and theoretically grounded aggregation beyond majority voting.
- We design *Confidence-Guided Early Stopping (CGES)*, which adaptively halts sampling to trade off accuracy and efficiency.
- We establish theoretical guarantees of correctness under ideal conditions where confidence scores are perfectly calibrated to the true probabilities of correctness, and extend the analysis to the realistic setting with noisy confidence estimates.
- We empirically validate CGES across five reasoning benchmarks, showing substantial efficiency improvements while maintaining or improving accuracy.

## 2 RELATED WORK

A widely used approach for test-time scaling is *self-consistency*, introduced by Wang et al. (2023), which aggregates multiple reasoning paths by majority vote to improve chain-of-thought reliability. However, it requires a fixed number of model calls and can fail when the correct answer is infrequent. To reduce this cost, Li et al. (2024) proposed early-stopping self-consistency (ESC), which stops sampling once predictions agree. More recent extensions, such as Self-Calibration (Huang et al., 2025), use dynamic stopping rules or distill self-consistency signals into single-pass confidence estimates. In contrast, our work introduces a **Bayesian framework** with **theoretical guarantees** for **confidence-guided early stopping**.

Beyond self-consistency, several methods adaptively allocate test-time compute. Snell et al. (2025) study compute-optimal scaling strategies, and inference scaling laws characterize how performance improves with additional sampling, guiding compute-efficient inference (Wu et al., 2025). Wang

et al. (2025) pose test-time scaling as optimal resource allocation, showing trajectory-level allocation can be suboptimal and introducing direction-oriented rollouts to reduce redundancy across semantically similar samples. Muennighoff et al. (2025) propose $s^1$, optimizing inference length under budget constraints. Other directions combine search and verification: self-enhanced tree search (Bi et al., 2025; Lample et al., 2022; Koh et al., 2024) expands multiple reasoning paths with sparse activation, while step-wise verifiers prune dynamically (Li et al., 2023; Lightman et al., 2024). Two-stage elimination approaches iteratively refine candidate answers (Chen et al., 2025), and query-variant ensembling improves robustness (Huang et al., 2024). These methods aim to balance accuracy and efficiency but differ in dependence on structured search, verifier signals, or semantic clustering. Our approach instead treats confidence as probabilistic evidence in a Bayesian model, enabling lightweight updates with formal consistency guarantees. Unlike methods focused on extracting a single answer or reasoning path, our framework samples multiple responses and then *aggregates* them via principled Bayesian inference.

A core component of our framework is *uncertainty estimation*, which has been studied extensively in generative LLMs. Probability-based methods such as length-normalized scoring (Malinin & Gales, 2021) mitigate biases toward shorter responses, while recent work incorporates meaning or trainable scoring. MARS (Bakman et al., 2024) adds semantics-aware token weighting, and LARS (Yaldiz et al., 2025) formulates supervised scoring over token-level features. Information-theoretic metrics distinguish epistemic and aleatoric uncertainty via iterative prompting (Abbasi-Yadkori et al., 2024). Similarity-based UQ methods such as SIMBA (Bhattacharjya et al., 2025) aggregate pairwise similarities for non-verbalized, largely black-box confidence estimates. Surveys (Geng et al., 2024) and unified taxonomies like the Uncertainty Estimation Codex (Xiao et al., 2025) provide overviews, and Bayesian or distillation-based approaches enable efficient estimation (Vejendla et al., 2025). Our goal is not to introduce new UQ methods; instead, we adapt existing techniques within the CGES framework, modifying them when necessary.

## 3 CONFIDENCE-BASED APPROACH

In this section, we propose a confidence-based method as an alternative to majority-vote self-consistency approach. We formalize the setting, introduce a Bayesian framework, provide theoretical guarantees under ideal conditions, and address realistic cases with noisy confidence scores.

### 3.1 PROBLEM SETTING AND NOTATIONS

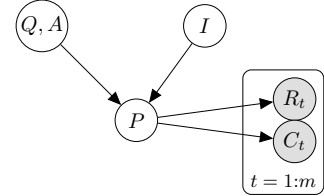

Suppose we query a large language model (LLM) multiple times with a given query $Q$, whose true answer is $A$, thereby obtaining a sample set $\mathcal{S}$ of responses. Let $\mathcal{U} = \{a_1, a_2, \ldots, a_K\}$ denote the complete set of all possible distinct candidate final answers from the LLM, where $K$ is the maximum number of such candidates. For the theoretical analysis here, we condition on the true answer $A$ appearing in the candidate set, i.e., there exists $I \in [K]$ with $a_I = A$, and use $I$ to denote the unknown correct index. If $A \notin \mathcal{U}$, no method operating only on $\{a_1, \ldots, a_K\}$ can recover $A$: it must abstain or pick an incorrect candidate. In such cases, CGES still returns the highest-posterior candidate in $\mathcal{U}$, but the consistency guarantees in Sections 3.4–3.5 do not apply. Conditioned on $(Q, A)$ and the identity of the correct candidate $I$, there exists a probability distribution over answers $P = (P_1, \ldots, P_K) \sim f_P(Q, A, I)$, where $P_j = \mathbb{P}[R = a_j \mid Q, A, I]$. Intuitively, this captures the stochastic behavior of the LLM given the query: the likelihood of producing each candidate answer depends both on the query and on which candidate is correct. For each LLM call $t = 1, 2, \ldots$, we draw a response–confidence pair $(R_t, C_t)$ according to $R_t \sim P$ and $C_t \sim f_{C|P}(P)$. Here $R_t$ denotes the *final textual answer string* produced by the LLM on call $t$ (e.g., "12", "$x = 3$", "yes"), after any post-processing or extraction step. Thus $R_t$ takes values in the finite candidate set $\mathcal{U}$, and is not the full sequence of generated tokens. The confidence signal $C_t$ is a scalar attached to $R_t$ and serves as a (possibly noisy) proxy for its probability of correctness. This structure can be represented as a graphical model, illustrated in Fig. 2.

Figure 2: Graphical model for the sampling process in our theoretical analysis.

**Idealistic vs. Realistic Assumptions.** It is useful to distinguish two types of assumptions. *Idealistic assumptions* are simplifying conditions enabling a clean, efficient algorithm and optimality

under ideal conditions (Theorem 1); these include a uniform error distribution and confidence scores independent of the true index. *Realistic assumptions* are expected to hold in practice and support both the CGES construction and guarantees under realistic conditions (Theorem 2); these include i.i.d. sampling assumption and a uniform prior on $I$. In the idealistic model, each $C_t$ is treated as an *exact* conditional correctness probability for $R_t$, encoded in the likelihood (1). Under realistic conditions, $C_t$ is only a *noisy estimator* of this probability, generated by some $f_{C|\mathbb{P}}$ (e.g., from token probabilities or a reward model), and Theorem 2 characterizes when consistency still holds. Thus, while all four assumptions 1–4 hold in the idealized setting, only the weaker pair 1–2 is retained in realistic regimes.

The four assumptions are as follows:

1. Given a fixed query $Q$, with a fixed $\mathcal{U}$ and $P$, the samples $\{(R_t, C_t)\}_{t=1}^m$ are i.i.d.
2. $I \sim \text{Uniform}(\{1, \ldots, K\})$. This prior over the *index* of the correct candidate treats all $K$ hypotheses symmetrically, reflecting arbitrary labeling of $\mathcal{U}$.
3. Under hypothesis $I = i$, the correct answer $a_i$ is emitted with probability $C_t$, while each incorrect candidate shares the residual probability mass uniformly:

$$\mathbb{P}(R_t = a_i \mid C_t, I = i) = C_t, \qquad \mathbb{P}(R_t = a_j \neq a_i \mid C_t, I = i) = \frac{1 - C_t}{K - 1}. \tag{1}$$

4. The confidence score $C_t$ is independent of the index of the correct answer. Intuitively, the confidence attached to a sample should not depend on which candidate happens to be correct. Formally, $\mathbb{P}(C_t \mid I = i) = \mathbb{P}(C_t \mid I = j) \quad \forall i, j \in \{1, \ldots, K\}$.

**Remark 1** (Interpretation of Assumptions 1–4). *For a fixed question $Q$, the LLM with a chosen confidence-estimation method induces a joint distribution over answer strings $R_t \in \mathcal{U}$ and confidences $C_t \in (0, 1)$. Assumption 1 treats repeated calls as i.i.d. draws from this distribution, a standard call-level abstraction in self-consistency and test-time scaling analyses (Snell et al., 2025; Wu et al., 2025) enabling the law-of-large-numbers arguments in Theorems 1 and 2. Assumption 2 imposes a symmetric prior over the index $I$ of the correct candidate, reflecting arbitrary labeling of $\mathcal{U}$. Assumption 3 specifies a one-versus-rest idealized likelihood parameterized by $C_t$, yielding a closed-form Bayesian update and the consistency result in Theorem 1. Assumption 4 requires that, before observing $R_t$, the marginal distribution of $C_t$ is invariant to relabelings of $\mathcal{U}$; it does* not *preclude higher $C_t$ on correct answers. Section 3.5 and Theorem 2 analyze a more realistic regime where data may deviate from this idealized likelihood, while CGES still uses it for scoring. Together, Assumptions 1–4 are standard modeling abstractions enabling clean, closed-form guarantees; the algorithm itself and the realistic analysis in Theorem 2 do not rely on these idealized conditions being exactly true in practice.*

Given these assumptions, the objectives of our framework are twofold: (i) to *identify the most probable index $i \in [K]$ corresponding to the true answer*, and (ii) to *quantify the level of confidence in this selection*.

## 3.2 BAYESIAN CONFIDENCE-BASED FRAMEWORK

We apply Bayesian inference to compute posterior probabilities. Given an unknown index $I \in \{1, \ldots, K\}$ and observed response–confidence pairs, the posterior distribution over $I$ is

$$\mathbb{P}(I = i \mid \text{Obs}) = \frac{\mathbb{P}(\text{Obs} \mid I = i)\, \mathbb{P}(I = i)}{\sum_{k=1}^{K} \mathbb{P}(\text{Obs} \mid I = k)\, \mathbb{P}(I = k)}.$$

Here, the numerator combines the likelihood of the observations under hypothesis $I = i$ and the prior $\mathbb{P}(I = i)$, while the denominator normalizes across all $K$ competing hypotheses. The set of observations is defined as $\text{Obs} = \{R_1, C_1, R_2, C_2, \ldots, R_m, C_m\}$, where $R_t$ denotes the response at step $t$ and $C_t$ its associated confidence score. This alternating structure reflects how, at each trial, both the raw prediction and its confidence are incorporated. The likelihood terms $\mathbb{P}(\text{Obs} \mid I = i)$ are therefore specified by our auxiliary generative model (cf. Section 3.1) and need not coincide with the true black-box LLM distribution $\mathbb{P}_{\text{LM}}(\cdot \mid Q)$. Since we assume a uniform prior over all hypotheses, i.e., $\mathbb{P}(I = i) = 1/K$, Bayes' rule simplifies to $\mathbb{P}(I = i \mid \text{Obs}) = \frac{\prod_t \mathbb{P}(R_t, C_t | I = i)}{\sum_{k=1}^{K} \prod_t \mathbb{P}(R_t, C_t | I = k)}$. This prior is purely a symmetric choice over the $K$ indices and is independent of the actual response distribution induced by the LLM. This formulation emphasizes that the posterior is built by multiplying the contributions of each observation and then renormalizing across all indices.

---

**Algorithm 1** SCORE: Confidence-Based Bayesian Normalization

---

**Require:** Candidate set $\mathcal{U} = \{a_1, \ldots, a_K\}$; samples $\mathcal{S} = \{(R_t, C_t)\}_{t=1}^m$ with $R_t \in \mathcal{U}$ and $C_t \in (0, 1)$

1: **for all** $a_i \in \mathcal{U}$ **do**
2: $\quad s_{a_i} \leftarrow \prod_{t:R_t = a_i} C_t \times \prod_{t:R_t \neq a_i} \frac{1-C_t}{K-1}$
3: **end for**
4: **return** $\mathrm{score}(a_i) = s_{a_i}/Z$ for all $a_i$

---

According to Assumption 4, we have $\mathbb{P}(C_t \mid I = i) = \mathbb{P}(C_t \mid I = j)$, $\forall i, j \in \{1, \ldots, K\}$. As a result, the joint distribution factorizes as $\mathbb{P}(R_t, C_t \mid I = i) = \mathbb{P}(R_t \mid C_t, I = i)\mathbb{P}(C_t)$, and the marginal $\mathbb{P}(C_t)$ cancels out in the numerator and denominator of Bayes' rule. This leads to the simplified posterior form in (2), where $X_i$ represents the posterior mass for hypothesis $i$.

$$\mathbb{P}(I = i \mid \mathrm{Obs}) \propto \frac{\prod_t \mathbb{P}(R_t \mid C_t, I = i)}{\sum_{k=1}^K \prod_t \mathbb{P}(R_t \mid C_t, I = k)} \triangleq X_i, \tag{2}$$

In the ideal scenario, when the response $R_t$ matches the true answer $a_i$, it is selected with probability equal to the reported confidence $C_t$, while the remaining $(K-1)$ incorrect answers share the residual probability mass uniformly. Formally:

$$\mathbb{P}(R_t \mid C_t, I = i) = \begin{cases} C_t & \text{if } R_t = a_i, \\ \frac{1-C_t}{K-1} & \text{otherwise.} \end{cases}$$

This ideal calibration assumption is used in Theorem 1; the realistic setting in Theorem 2 relaxes it and treats $C_t$ as a noisy estimator of this probability.

### 3.3 ALGORITHM FOR CONFIDENCE-BASED SCORING

Given sampled answers and their confidences $\mathcal{S} = \{(R_t, C_t)\}_{t=1}^m$ for a single question, our goal is to convert them into calibrated, comparable probabilities over the unique answer set $\mathcal{U} = \{a_t\}_{t=1}^K$. The Bayesian posterior of Eq 2 factorizes into a product of per-sample terms: if the hypothesis (answer) is $a_i$, then a sample $R_t$ that outputs $a_i$ has likelihood $C_t$, and any other answer has likelihood $(1 - C_t)/(K - 1)$. We therefore form an unnormalized score $s_{a_i}$ by multiplying these terms across all samples and then normalize across candidates. Algorithm 1 implements this computation.

Algorithm 2 wraps SCORE into an adaptive loop that allocates test-time compute per question. We begin with one sample per question, compute posteriors with SCORE, and maintain the set of unresolved questions $D_{\mathrm{rem}}$ whose current top posterior is below a confidence threshold $\gamma$. At round $t = 2, \ldots, B$, we query the LLM *only* for $n \in D_{\mathrm{rem}}$, append the new $(R_t^n, C_t^n)$, recompute SCORE on that question's $t$ samples, and remove it from $D_{\mathrm{rem}}$ as soon as its top posterior exceeds $\gamma$. The process stops when all questions are confident or the budget $B$ is reached, returning the argmax label per question and the average number of LLM calls. If no candidate ever exceeds the threshold $\gamma$ for a given question before the budget $B$ is exhausted, CGES still returns the current argmax posterior label for that question. Thus, $\gamma$ controls test-time compute per question rather than enforcing abstention; extending the framework with a "no-answer" option is left for future work.

### 3.4 THEORETICAL ANALYSIS OF ALGORITHM PERFORMANCE (IDEAL SCENARIO)

**Theorem 1.** *Under Assumptions 1, 2, 3, and 4, and provided the confidences are informative i.e., $\mathbb{P}(C_t = 1/K) = 0$, the Bayesian confidence-based aggregator identifies the correct answer with probability tending to one as the number of samples $m$ grows:*

$$\mathbb{P}\left(\arg\max_{i \in [K]} X_i = I\right) \longrightarrow 1 \quad as \quad m \to \infty.$$

*In fact, $X_I \to 1$ and $X_k \to 0$ for all $k \neq I$ almost surely.*

**Proof Sketch.** Fix any wrong index $k \neq I$ and compare the (log) likelihood of the observed samples under the hypotheses $I$ vs. $k$. Each sample contributes a log-likelihood ratio (LLR) increment whose *expected* value is strictly positive whenever the confidence $C_t$ deviates from the uninformative value

---

**Algorithm 2** Confidence-Guided Early Stopping (CGES)

---

**Require:** $N$ questions; threshold $\gamma$; calls budget $B$; scoring routine SCORE
1: Initialize $\text{scores}_n \leftarrow \text{SCORE}(\{(R_1^n, C_1^n)\})$ for all $n \in [N]$
2: $D_{\text{rem}} \leftarrow [N]$; calls $\leftarrow N$
3: **for** $t = 2$ to $B$ **do**
4:      $D_{\text{rem}} \leftarrow \{ n \in D_{\text{rem}} : \max_i \text{scores}_n[i] < \gamma \}$
5:      **if** $D_{\text{rem}} = \emptyset$ **then break**
6:      **end if**
7:      Query LLM for each $n \in D_{\text{rem}}$; calls $+= |D_{\text{rem}}|$
8:      **for all** $n \in D_{\text{rem}}$ **do**
9:          $\text{scores}_n \leftarrow \text{SCORE}(\{(R_1^n, C_1^n), \ldots, (R_t^n, C_t^n)\})$
10:     **end for**
11: **end for**
12: **return** $\hat{y}_n = a_{\arg \max_i \text{scores}_n[i]} \; \forall n$, usage$=$ calls$/N$

---

$1/K$. By the Strong Law of Large Numbers (SLLN), these positive-drift increments accumulate linearly, so the total LLR diverges to $+\infty$. Hence the likelihood under the true index dominates every competitor, forcing the normalized posterior $X_I$ to 1 and all others to 0. A complete formal proof is provided in Appendix A

### 3.5 THEORETICAL ANALYSIS OF ALGORITHM PERFORMANCE (REALISTIC SCENARIO)

In contrast to the ideal case, where the data are generated by the same likelihood used by the aggregator, the realistic setting permits model mismatch: the observed answers $R_t$ are drawn from an unknown but fixed (per question) distribution $\mathbf{P} = (P_1, \ldots, P_K)$, while the confidence signal $C_t$ is a noisy proxy produced by an estimator (for example token probabilities). The aggregator itself retains the same one-versus-rest likelihood as in the ideal model; under these conditions, consistency reduces to the sign of the average LLR drift $\mu_k$ defined below.

**Theorem 2** (Consistency under realistic confidence noise). *Assume 1 and 2. For a fixed question, let $R_t \sim \mathbf{P} = (P_1, \ldots, P_K)$ i.i.d., and $C_t \mid \mathbf{P} \sim f_{C|\mathbf{P}}(\cdot \mid \mathbf{P})$ i.i.d., with $C_t \in (0,1)$ a.s. and $\mathbb{E}[|\log C_t| + |\log(1 - C_t)|] < \infty$. The aggregator uses the one–versus–rest likelihood from the ideal model. Let $\theta_t = (1 - C_t)/(K - 1)$ and define*

$$\mu_k = \mathbb{E}_{\mathbf{P},C}\left[(P_1 - P_k) \log\left(\frac{C_t}{\theta_t}\right)\right], \qquad k \neq 1.$$

*If $\mu_k > 0$ for all $k \neq 1$, then $\mathbb{P}(\arg \max_{i \in [K]} X_i = 1) \to 1$ as $m \to \infty$ and, in fact, $X_1 \to 1$ and $X_k \to 0$ almost surely. If $\mu_{k^\star} < 0$ for some $k^\star \neq 1$, then $X_1 \to 0$ almost surely.*

**Proof Sketch.** As in the ideal case, fix any $k \neq 1$ and consider the log-likelihood ratio (LLR) between hypotheses $I = 1$ and $I = k$. Under the realistic generator, $(R_t, C_t)$ are drawn with $R_t \sim \mathbf{P}$ while the aggregator evaluates likelihoods using $C_t$ and $\theta_t = (1 - C_t)/(K - 1)$. The per-sample LLR increment has conditional mean $\mathbb{E}[Y_k^{(t)} \mid \mathbf{P}, C_t] = (P_1 - P_k)(\log C_t - \log \theta_t)$. Averaging over $(\mathbf{P}, C_t)$ gives the drift $\mu_k$. If $\mu_k > 0$, the Strong Law of Large Numbers implies the cumulative LLR grows linearly to $+\infty$, so the likelihood under $I = 1$ dominates and the posterior concentrates on the truth. This condition also covers minority-correct regimes ($P_1 < P_k$) provided $C_t$ is systematically below $1/K$, which flips the sign of the log term and yields positive drift where majority vote would fail. A full proof appears in Appendix B.

**Remark 2** (Missing ground-truth among candidates). *Both Theorem 1 and Theorem 2 analyze the behavior of CGES under the event that the true answer $A$ is contained in the candidate set $\mathcal{U}$, so that $I$ is well-defined. If $A \notin \mathcal{U}$ for a given question, then any aggregation procedure that operates only on $\mathcal{U}$ inevitably fails to return $A$: the error probability has an irreducible component $\mathbb{P}(A \notin \mathcal{U})$ that no self-consistency-style scheme can eliminate. In this misspecified regime, CGES behaves like other sampling-based methods: it selects the candidate in $\mathcal{U}$ with the highest posterior mass, and the consistency guarantees above no longer hold. In practice, a high threshold $\gamma$ can be used to signal such high-uncertainty cases (e.g., when no candidate accumulates substantial posterior probability), but formal guarantees on correctness are impossible without access to the true answer.*

## 4 EXPERIMENTS

We evaluate CGES on five reasoning benchmarks using two 7B-class models and compare against self-consistency (SC) (Wang et al., 2023), early-stopping self-consistency (ESC) (Li et al., 2024), and Adaptive-Consistency (ASC) (Aggarwal et al., 2023). *SC* aggregates multiple samples by majority vote. *ESC* halts once recent predictions align, reducing calls. *ASC* stops dynamically when the estimated majority is stable. We report *accuracy* of the final answer and *efficiency* as the average number of LLM calls per question. Results are averaged over three seeds. Decoding and prompting follow prior work (Appendix D). Confidence signals $C_t$ use the strategies in Section 4.2.

### 4.1 DATASETS AND MODELS

We evaluate on five benchmarks spanning mathematics and broad knowledge: **AIME24**, 30 problems from the 2024 American Invitational Mathematics Examination; **MATH500**, a 500-question subset of MATH (Hendrycks et al., 2021) targeting advanced reasoning; **GSM8K** (Cobbe et al., 2021), 8,500 grade-school math word problems; **MMLU_Pro** (Wang et al., 2024), 14 college-level subjects; and **GPQA Diamond** (Rein et al., 2024), expert-written science questions challenging even for skilled humans. For the harder datasets (AIME24, GPQA), we use *DeepSeek-R1-Distill-Qwen(7B)* (DeepSeek-AI et al., 2025), and for the easier ones (MATH500, GSM8K, MMLU_Pro) we use *Qwen2.5(7B)* (Yang et al., 2025).

### 4.2 CONFIDENCE ESTIMATION STRATEGIES

We compare several strategies for estimating the *scalar* confidence $C_t \in (0,1)$ of each sampled answer $R_t$. Each strategy maps model outputs (token probabilities or reward scores) to a scalar $C_t$ that we interpret as an estimate of the probability that the corresponding response $R_t$ is correct. These estimates need not be perfectly calibrated; they serve as noisy confidence signals in the sense formalized in Section 3.1 and Theorem 2. Our framework operates at the response level, but confidence estimation relies on finer token-level granularities. Specifically, let a response $R_t$ consist of a token sequence $T_1, \ldots, T_L$ with associated autoregressive probabilities $p_1, \ldots, p_L$. We use lowercase $p_\ell$ to denote these token-level probabilities from the underlying language model, and uppercase $\mathbb{P}(\cdot)$ for probabilities over candidate answers $a_j \in \mathcal{U}$ as in Section 3.1. The role of the present section is to map token-level scores (or reward-model outputs) into a single scalar confidence $C_t$ Building on this, we now describe three token-based approaches and one verifier-based alternative.

**Length-Normalized Scoring (LNS)** (Malinin & Gales, 2021). A natural way to quantify the likelihood of a response is by averaging over token probabilities. The *geometric mean* yields the standard length-normalized score $\text{LNS}_{\text{geom}} = \exp\left(\frac{1}{L}\sum_{\ell=1}^{L}\log p_\ell\right)$ while the *arithmetic mean* provides a simpler length-insensitive proxy, $\text{LNS}_{\text{arith}} = \frac{1}{L}\sum_{\ell=1}^{L} p_\ell$. We set $C_t$ to either of these values and denote them in results as *LNS [Geometric mean]* and *LNS [Arithmetic mean]*.

**MARS (Step-Weighted Scoring)** (Bakman et al., 2024). The MARS method generalizes LNS by assigning different weights to different positions in the sequence. Each token $T_\ell$ receives an exponent $w(R, Q, L, \ell) \triangleq \frac{1}{2L} + \frac{u(R,Q,\ell)}{2}$, where $R$ denotes the full textual response string (suppressing the time index $t$ for brevity), so the overall score becomes $\bar{P}(R \mid Q, \theta) = \prod_{\ell=1}^{L} p_\ell^{w(R,Q,L,\ell)}$ where $\theta$ denotes the parameters of the language model generating token probabilities. Here $u(R, Q, \ell)$ is an importance score for token $T_\ell$, such as the semantic change in the output when masking that token. While token-level weighting can be precise, for long reasoning responses most token importance scores become nearly uniform, and computing $u(\cdot)$ for every token is expensive (requiring $L$ calls to a semantic extractor model such as a sentence transformer). To address this, we adopt a *step-wise* variant of MARS: instead of per-token weights, we assign weights at the granularity of reasoning steps or sentence segments. Although the step-importance score is recomputed each iteration, this overhead (6 layers, ~50M parameters) is negligible compared to the 7B-parameter inference model we query for $R_t$. We denote the resulting confidence as $C_t = \text{MARS}$.

**Reward Model Confidence.** In addition to the above token-level methods, we also consider a model-based approach that evaluates entire responses directly. A trained reward model assigns a quality score to each $R_t$, which we use as $C_t$. In our study, we use `Qwen2.5-Math-PRM-72B` process reward model (Zhang et al., 2025), which outputs a scalar in $(0,1)$ that correlates with

Table 1: Accuracy (%) and avg. #Calls across five reasoning tasks; Avg is mean over benchmarks. Parentheses show difference vs. SC (#Calls=16 or SC Acc). For CGES, the first row corresponds to the *Efficient* setting (lower threshold, fewer calls) and the second row to the *Conservative* setting (higher threshold, more calls).[1]

| | AIME24 | | MATH500 | | GSM8K | | GPQA | | MMLU_Pro | | Avg. | |
|---|---|---|---|---|---|---|---|---|---|---|---|---|
| | #Calls | Acc | #Calls | Acc | #Calls | Acc | #Calls | Acc | #Calls | Acc | #Calls | Acc |
| **SC** | 16.00 | 78.89 | 16.00 | 82.20 | 16.00 | 94.39 | 16.00 | 51.01 | 16.00 | 61.54 | 16.00 | 73.61 |
| **ESC (w=4)** | 11.02 | 78.89 | 7.88 | 82.07 | 5.22 | 94.37 | 11.90 | 51.18 | 9.04 | 61.43 | 9.01 | 73.59 |
| | (-4.98) | (+0.00) | (-8.12) | (-0.13) | (-10.78) | (-0.02) | (-4.10) | (+0.17) | (-6.96) | (-0.11) | (-6.99) | (-0.02) |
| **ESC (w=8)** | 14.40 | 78.89 | 11.57 | 82.20 | 9.50 | 94.39 | 14.87 | 51.01 | 12.94 | 61.54 | 12.66 | 73.61 |
| | (-1.60) | (+0.00) | (-4.43) | (+0.00) | (-6.50) | (+0.00) | (-1.13) | (+0.00) | (-3.06) | (+0.00) | (-3.34) | (+0.00) |
| **Adaptive-Consistency (BETA)** | 10.25 | 78.89 | 7.36 | 82.07 | 5.05 | 94.37 | 11.43 | 50.51 | 8.58 | 61.46 | 8.53 | 73.46 |
| | (-5.75) | (+0.00) | (-8.64) | (-0.13) | (-10.95) | (-0.02) | (-4.57) | (+0.50) | (-7.42) | (-0.08) | (-7.47) | (-0.15) |
| **Adaptive-Consistency (Dirichlet)** | 11.17 | 78.89 | 7.88 | 82.07 | 5.22 | 94.37 | 12.18 | 50.51 | 9.13 | 61.46 | 9.12 | 73.46 |
| | (-4.83) | (+0.00) | (-8.12) | (-0.13) | (-10.78) | (-0.02) | (-3.82) | (+0.50) | (-6.87) | (-0.08) | (-6.88) | (-0.15) |
| **CGES (Ours)** | | | | | | | | | | | | |
| **LNS[Arithmetic mean]** | 6.47 | 78.89 | 4.69 | 81.93 | 4.50 | 94.26 | **2.09** | 51.13 | 6.77 | 61.56 | **4.90** | 73.55 |
| | (-9.53) | (+0.00) | (-11.31) | (-0.27) | (-11.50) | (-0.13) | (-13.91) | (+0.12) | (-9.23) | (+0.02) | (-11.10) | (-0.06) |
| | 9.59 | 78.89 | 5.81 | 81.87 | 4.50 | 94.26 | 10.48 | 51.52 | 8.29 | 61.58 | 7.73 | **73.62** |
| | (-6.41) | (+0.00) | (-10.19) | (-0.33) | (-11.50) | (-0.13) | (-5.52) | (+0.51) | (-7.71) | (+0.04) | (-8.27) | (+0.01) |
| **LNS[Geometric mean]** | 7.27 | 78.44 | 6.64 | 82.00 | 5.36 | 94.36 | 4.44 | 51.35 | 6.78 | 61.63 | 6.88 | 73.56 |
| | (-8.73) | (-0.45) | (-9.36) | (-0.20) | (-10.64) | (-0.03) | (-11.56) | (+0.34) | (-9.22) | (+0.09) | (-9.12) | (-0.05) |
| | 11.56 | 78.44 | 6.64 | 82.00 | 5.36 | 94.36 | 13.26 | 51.18 | 10.68 | 61.65 | 9.90 | 73.52 |
| | (-4.44) | (-0.45) | (-9.36) | (-0.20) | (-10.64) | (-0.03) | (-2.74) | (+0.17) | (-5.32) | (+0.11) | (-6.10) | (-0.09) |
| **MARS** | **5.79** | 77.78 | 6.80 | 81.93 | 5.39 | **94.42** | 3.61 | **52.69** | 6.68 | 61.60 | 5.65 | 73.28 |
| | (-10.21) | (-1.11) | (-9.20) | (-0.27) | (-10.61) | (+0.03) | (-12.39) | (+1.68) | (-9.32) | (+0.06) | (-10.35) | (-0.33) |
| | 10.83 | 77.78 | 6.80 | 81.93 | 5.39 | 94.42 | 12.14 | 50.84 | 10.59 | 61.53 | 9.15 | 73.52 |
| | (-5.17) | (-1.11) | (-9.20) | (-0.27) | (-10.61) | (+0.03) | (-3.86) | (-0.17) | (-5.41) | (-0.01) | (-6.85) | (-0.09) |

Table 2: **CGES–PRM (upper bound).** Same conventions as Table 1. Confidences from a large PRM (`Qwen2.5-Math-PRM-72B`; scoring only).

| | AIME24 | | MATH500 | | GSM8K | | GPQA | | MMLU_Pro | | Avg. | |
|---|---|---|---|---|---|---|---|---|---|---|---|---|
| | #Calls | Acc | #Calls | Acc | #Calls | Acc | #Calls | Acc | #Calls | Acc | #Calls | Acc |
| **RM Confidence** | 6.71 | 77.78 | 4.32 | 83.00 | 2.57 | 94.49 | 6.62 | 51.68 | 6.05 | 62.17 | 5.27 | 73.42 |
| | (-9.29) | (-1.11) | (-11.68) | (+0.80) | (-13.43) | (+0.10) | (-9.38) | (+0.67) | (-9.95) | (+0.63) | (-10.73) | (-0.19) |
| | 7.65 | 77.78 | 5.27 | 85.13 | 3.02 | 95.55 | 10.62 | 50.67 | 7.44 | 63.64 | 6.40 | 74.35 |
| | (-8.36) | (-1.11) | (-10.73) | (+2.93) | (-12.98) | (+1.16) | (-5.38) | (-0.34) | (-8.57) | (+2.10) | (-9.60) | (+0.74) |

alignment to the ground truth on math-style reasoning. Because it has 72B parameters (far larger than our 7B inference model), using it as the scorer is *impractical* for deployment; we include it as a near-optimal reference to approximate an upper bound on confidence quality, especially on tasks that are in-domain for this PRM. We denote this variant as *RM Confidence*.

## 4.3 RESULTS

Table 1 reports accuracy and average number of calls for CGES and baselines with a Self-Consistency budget of $B = 16$. We compare probability-based variants of CGES, which use token-level (LNS) and step-level (MARS) confidence scores. Across all benchmarks, CGES variants using token- and step-level confidence (LNS and MARS) markedly reduce model calls compared to SC and ESC while maintaining near-identical accuracy. With the strongest efficiency gains, arithmetic-mean LNS averages **4.90 calls** (**69.4%** fewer than SC's 16) with only $-0.06\%$ accuracy change. MARS further improves efficiency on harder tasks such as GPQA, lowering calls from **16.00 to 3.61** with a **+1.68%** accuracy gain. On MMLU_Pro, CGES attains comparable or better accuracy with fewer than half the calls (e.g., 6.77 vs. 16.00 with $+0.02\%$). These results show CGES outperforms local majority-vote stopping rules, enabling adaptive early stopping from confidence signals. For easier tasks (MATH500, GSM8K), probability-based variants show minor accuracy dips despite large call savings, as improvements then depend on *calibration* of $C_t$; noisy probability proxies (LNS/MARS) can be slightly over- or under-confident, limiting accuracy gains. Even so, efficiency gains remain substantial (often $> 2\times$ fewer calls) with accuracy effectively preserved.

---

[1]The *Efficient* setting corresponds to the smallest $\gamma$ for which CGES matches or surpasses SC performance, while the *Conservative* setting corresponds to the largest $\gamma$ considered. If CGES does not reach SC performance, both settings coincide at the largest $\gamma$.

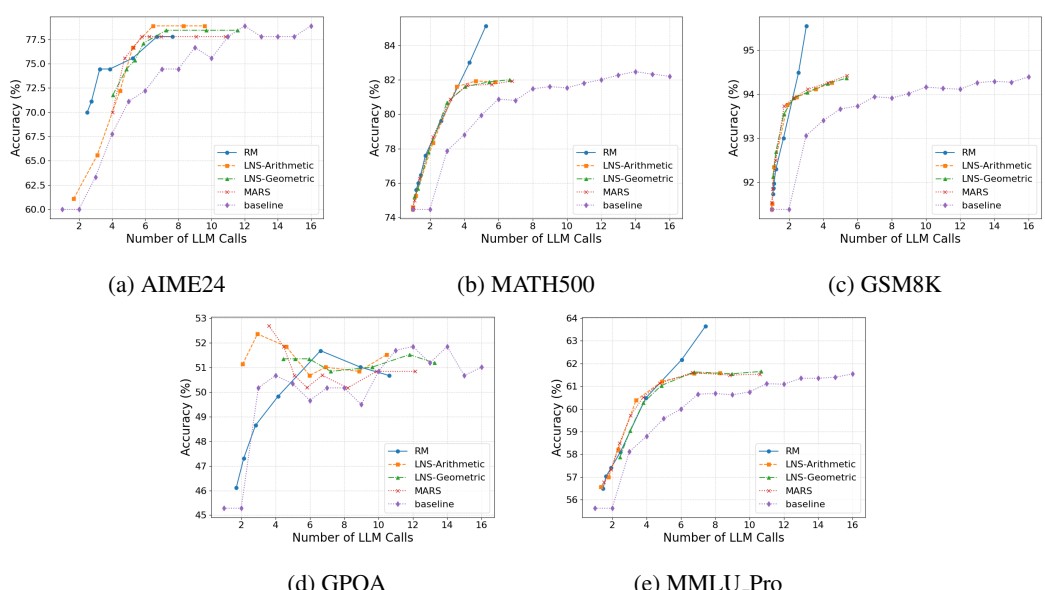

Figure 3: Accuracy vs. number of LLM calls ($B$=16) on AIME24 (a), MATH500 (b), GSM8K (c), GPQA (d), and MMLU_Pro (e). CGES achieves near-maximal accuracy with far fewer calls than self-consistency.

Reward-model–based CGES approximates a best-case scenario where confidence estimates are closer to ground truth than those from the 7B inference models. Since `Qwen2.5-Math-PRM-72B` is larger and trained for step-level scoring, its signals are stronger than what is practical at inference time. Improvements are most pronounced on math-centric datasets (MATH500, GSM8K), where the PRM's training overlaps with the task: accuracy increases while calls drop sharply (Table 2). On AIME24 and GPQA, domain mismatch and higher difficulty limit PRM utility, yielding small accuracy drops despite fewer calls. On MMLU_Pro, the PRM still provides useful confidence signals and surpasses SC with substantially fewer calls. Appendix C reports additional results under smaller SC budgets ($B = 4, 8$), showing consistent trends, as well as calibration analyses and large-model (70B–72B) experiments.

Figure 3 shows the accuracy–efficiency trade-off. Each CGES curve is obtained by sweeping the stopping threshold $\gamma$ (from 0.7 to 0.9999); the *baseline* traces self-consistency (SC) at fixed budgets. On AIME24, GSM8K, and MATH500, CGES achieves near-maximal accuracy after only a few calls, whereas SC requires the full budget. **Reward-model (PRM) confidence** converges fastest (often within 3–4 calls), and is shown as a near-ideal reference rather than a deployable setting. Across datasets, curves flatten beyond ∼6 calls, indicating diminishing returns and that SC's $B$=16 is over-provisioned. These results confirm that confidence-guided stopping enables CGES to adaptively terminate sampling early without compromising accuracy.

## 5 CONCLUSIONS

We proposed CGES, a confidence-based Bayesian framework for test-time scaling of LLMs. By treating each response and its confidence as probabilistic evidence, CGES enables early stopping and more reliable aggregation than majority voting. Across five reasoning benchmarks, CGES substantially reduces model calls while maintaining or improving accuracy, outperforming Self-Consistency and early-stopping baselines. Our theoretical analysis further shows correctness under both ideal and noisy confidence assumptions. Overall, confidence-guided aggregation provides a principled solution within the family of test-time scaling methods that sample multiple responses and aggregate them into a single answer. Future work includes (i) developing more accurate confidence estimators to further enhance efficiency and accuracy, and (ii) predicting the required number of samples dynamically from confidence signals.

## 6 REPRODUCIBILITY STATEMENT

We have taken several steps to ensure the reproducibility of our work. All theoretical claims are stated under clearly enumerated assumptions (Section 3.1) with full proofs provided in Appendices A–B. The complete Bayesian formulation and algorithmic details of CGES are presented in Section 3.2–3.3, including pseudocode for both the scoring and stopping procedures (Algorithms 1-2). Experimental protocols are fully described in Section 4, covering datasets (Section 4.1), models, baselines, and confidence estimation strategies (Section 4.2). Detailed hyperparameters, decoding configurations, and additional results under varying budgets are included in Appendix C-D. To further facilitate verification, we provide an anonymized implementation and experiment scripts as supplementary material. Together, these resources ensure that both the theoretical and empirical results reported in this paper can be independently reproduced and validated.

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

## A  FULL PROOF OF THEOREM 1

*Proof.* Without loss of generality, relabel so that the true index is $I = 1$. Define the unnormalized and normalized posteriors

$$A_j := \prod_{t=1}^m \mathbb{P}(R_t \mid C_t, I = j), \qquad X_j := \frac{A_j}{\sum_{k=1}^K A_k}, \qquad j \in [K].$$

To prove the claim, it is enough to show that for every $k \neq 1$, $A_1/A_k \to \infty$ almost surely, which then implies $X_1 \to 1$ and $X_k \to 0$ almost surely.

**Step 1:**  First, for a fixed $k \neq 1$, we define

$$Y_k^{(t)} := \log \mathbb{P}(R_t \mid C_t, I = 1) - \log \mathbb{P}(R_t \mid C_t, I = k).$$

Under Assumption 3, conditioning on $C_t$ we have

$$\log \mathbb{P}(R_t \mid C_t, I = 1) = \begin{cases} \log C_t & \text{if } R_t = a_1, \\ \log\left(\frac{1-C_t}{K-1}\right) & \text{if } R_t \neq a_1, \end{cases}$$

and

$$\log \mathbb{P}(R_t \mid C_t, I = k) = \begin{cases} \log C_t & \text{if } R_t = a_k, \\ \log\left(\frac{1-C_t}{K-1}\right) & \text{if } R_t \neq a_k. \end{cases}$$

Let $\theta_t := \frac{1-C_t}{K-1}$. Using Assumption 1 (i.i.d. across $t$ given $P$ and hence given $C_t$ in this idealized parameterization), the conditional probabilities under the true model $I = 1$ are $\mathbb{P}(R_t = 1 \mid C_t, I = 1) = C_t$ and $\mathbb{P}(R_t = k \mid C_t, I = 1) = \theta_t$.

**Step 2:**  In the next step, by taking the conditional expectation of $Y_k^{(t)}$ given $C_t$, it follows that

$$\mathbb{E}\left[Y_k^{(t)} \mid C_t\right] = \left(C_t - \theta_t\right)\left(\log C_t - \log \theta_t\right) = \left(C_t - \theta_t\right)\log\left(\frac{C_t}{\theta_t}\right).$$

Because $x \mapsto \log x$ is strictly increasing, $(a - b)\log(a/b) > 0$ for $a \neq b$. Here $a = C_t$ and $b = \theta_t$, so the conditional mean is strictly positive whenever $C_t \neq \theta_t$, i.e., whenever $C_t \neq 1/K$. By the informativeness condition $\mathbb{P}(C_t = 1/K) = 0$,

$$\mu_k := \mathbb{E}\left[Y_k^{(t)}\right] = \mathbb{E}\left[\mathbb{E}\left[Y_k^{(t)} \mid C_t\right]\right] > 0.$$

Moreover, $|Y_k^{(t)}|$ has finite expectation since $C_t \in (0, 1)$ a.s., making the logs finite.

**Step 3:**  By Assumption 1, $\{Y_k^{(t)}\}_{t=1}^m$ are i.i.d. with $\mathbb{E}[|Y_k^{(t)}|] < \infty$ and $\mathbb{E}[Y_k^{(t)}] = \mu_k > 0$. Finally, the Strong Law of Large Numbers yields

$$\frac{1}{m}\sum_{t=1}^m Y_k^{(t)} \xrightarrow{\text{a.s.}} \mu_k > 0 \implies \sum_{t=1}^m Y_k^{(t)} \xrightarrow{\text{a.s.}} +\infty.$$

Exponentiating both sides of $\log(A_1/A_k) = \sum_{t=1}^{m} Y_k^{(t)}$ gives

$$\frac{A_1}{A_k} \;=\; \exp\Big(\sum_{t=1}^{m} Y_k^{(t)}\Big) \xrightarrow{\text{a.s.}} \infty.$$

Since this holds for every $k \neq 1$, we have $A_k/A_1 \to 0$ almost surely for all $k \neq 1$, and therefore

$$X_1 \;=\; \frac{1}{1 + \sum_{k \neq 1} A_k/A_1} \xrightarrow{\text{a.s.}} 1, \qquad X_k \;=\; \frac{A_k/A_1}{1 + \sum_{j \neq 1} A_j/A_1} \xrightarrow{\text{a.s.}} 0.$$

This completes the proof. $\qquad\square$

## B  FULL PROOF OF THEOREM 2

*Proof.* Without loss of generality, let $I = 1$. For $j \in [K]$, define

$$A_j \;:=\; \prod_{t=1}^{m} \mathbb{P}(R_t \mid C_t, I = j), \qquad X_j \;:=\; \frac{A_j}{\sum_{k=1}^{K} A_k}.$$

It suffices to show that for every $k \neq 1$, $A_1/A_k \xrightarrow{\text{a.s.}} \infty$, which implies $X_1 \to 1$ and $X_k \to 0$ almost surely.

**Step 1:**  In the first step, we fix $k \neq 1$ and set

$$Y_k^{(t)} \;:=\; \log \mathbb{P}(R_t \mid C_t, I = 1) \;-\; \log \mathbb{P}(R_t \mid C_t, I = k).$$

With the model likelihood above and $\theta_t = \frac{1 - C_t}{K - 1}$,

$$\log \mathbb{P}(R_t \mid C_t, I = 1) = \begin{cases} \log C_t & (R_t = a_1), \\ \log \theta_t & (R_t \neq a_1), \end{cases} \qquad \log \mathbb{P}(R_t \mid C_t, I = k) = \begin{cases} \log C_t & (R_t = a_k), \\ \log \theta_t & (R_t \neq a_k). \end{cases}$$

**Step 2:**  Next by conditioning on $(\mathbf{P}, C_t)$ and using $\mathbb{P}(R_t = a_r \mid \mathbf{P}) = P_r$, we have

$$\mathbb{E}\Big[Y_k^{(t)} \mid \mathbf{P}, C_t\Big] = P_1(\log C_t - \log \theta_t) + P_k(\log \theta_t - \log C_t) = (P_1 - P_k)\big(\log C_t - \log \theta_t\big).$$

Taking expectations over $(\mathbf{P}, C_t)$ gives

$$\mu_k \;:=\; \mathbb{E}\Big[Y_k^{(t)}\Big] = \mathbb{E}_{\mathbf{P}, C}\Big[(P_1 - P_k)\log\Big(\tfrac{C_t}{\theta_t}\Big)\Big].$$

By assumption, $\mu_k > 0$ for all $k \neq 1$ and $\mathbb{E}[|Y_k^{(t)}|] < \infty$ (since $C_t \in (0, 1)$ a.s. and the logs are integrable).

**Step 3:**  By Assumption 1 and the i.i.d. generation of $(R_t, C_t)$ given $\mathbf{P}$, the increments $\{Y_k^{(t)}\}_{t=1}^{m}$ are i.i.d. with finite first moment and mean $\mu_k > 0$. The Strong Law of Large Numbers yields

$$\frac{1}{m} \sum_{t=1}^{m} Y_k^{(t)} \to \mu_k \;>\; 0 \text{ almost surely} \quad \Longrightarrow \quad \sum_{t=1}^{m} Y_k^{(t)} \to +\infty \text{ almost surely}.$$

Hence

$$\frac{A_1}{A_k} = \exp\Big(\sum_{t=1}^{m} Y_k^{(t)}\Big) \to \infty \text{ almost surely},$$

so $A_k/A_1 \to 0$ a.s. for all $k \neq 1$, and therefore $X_1 \to 1$ and $X_k \to 0$ almost surely.

**Converse (necessity).**  If $\mu_{k^\star} < 0$ for some $k^\star \neq 1$, then by the same SLLN argument, $\sum_{t=1}^{m} Y_{k^\star}^{(t)} \to -\infty$ a.s., so $A_1/A_{k^\star} \to 0$ almost surely and thus $X_1 \to 0$ almost surely. (When $\mu_k = 0$ for some $k$, the LLR has zero drift and the posterior need not concentrate; this is a boundary case.) $\qquad\square$

## C ADDITIONAL RESULTS

Additional results are provided in Table 3 and Table 4.

| | AIME24 #Calls | AIME24 Acc | MATH500 #Calls | MATH500 Acc | GSM8K #Calls | GSM8K Acc | GPQA #Calls | GPQA Acc | MMLU_Pro #Calls | MMLU_Pro Acc | Avg. #Calls | Avg. Acc |
|---|---|---|---|---|---|---|---|---|---|---|---|---|
| **SC** | 4.00 | 67.78 | 4.00 | 78.80 | 4.00 | 93.40 | 4.00 | 50.67 | 4.00 | 58.79 | 4.00 | 69.89 |
| **CGES (Ours)** | | | | | | | | | | | | |
| **LNS[Arithmetic mean]** | 3.03 | 68.89 | 2.59 | 79.40 | 1.84 | 93.48 | 1.52 | 53.02 | 2.76 | 59.05 | 2.35 | 70.77 |
| | (–0.97) | (+1.11) | (–1.41) | (+0.60) | (–2.16) | (+0.08) | (–2.48) | (+2.35) | (–1.24) | (+0.26) | (–1.65) | (+0.88) |
| | 4.00 | 68.89 | 3.75 | 79.40 | 3.86 | 93.48 | 4.00 | 53.02 | 3.98 | 59.05 | 3.92 | 70.77 |
| | (0.00) | (+1.11) | (–0.25) | (+0.60) | (–0.14) | (+0.08) | (0.00) | (+2.35) | (–0.02) | (+0.26) | (–0.08) | (+0.88) |
| **LNS[Geometric mean]** | 3.17 | 67.99 | 2.93 | 79.40 | 2.14 | 93.40 | 3.33 | 53.54 | 3.02 | 59.00 | 2.92 | 70.67 |
| | (–0.83) | (+0.21) | (–1.07) | (+0.60) | (–1.86) | (+0.00) | (–0.68) | (+2.87) | (–0.98) | (+0.21) | (–1.08) | (+0.78) |
| | 4.00 | 67.78 | 3.95 | 79.47 | 3.99 | 93.35 | 4.00 | 53.20 | 4.00 | 59.34 | 3.99 | 70.63 |
| | (0.00) | (+0.00) | (–0.05) | (+0.67) | (–0.01) | (–0.05) | (0.00) | (+2.53) | (0.00) | (+0.55) | (–0.01) | (+0.74) |
| **MARS** | 2.48 | 67.78 | 3.00 | 79.00 | 2.15 | 93.45 | 3.00 | 53.05 | 3.05 | 59.06 | 2.74 | 70.47 |
| | (–1.52) | (+0.00) | (–1.00) | (+0.20) | (–1.85) | (+0.05) | (–1.00) | (+2.38) | (–0.95) | (+0.27) | (–1.26) | (+0.58) |
| | 4.00 | 72.22 | 3.98 | 79.07 | 3.99 | 93.38 | 4.00 | 51.85 | 4.00 | 59.14 | 3.99 | 71.13 |
| | (0.00) | (+4.44) | (–0.02) | (+0.27) | (–0.01) | (–0.02) | (0.00) | (+1.18) | (0.00) | (+0.35) | (–0.01) | (+1.24) |
| **CGES - Near Ideal Scenario (Ours)** | | | | | | | | | | | | |
| **RM Confidence** | 1.72 | 68.89 | 1.80 | 78.93 | 2.16 | 94.16 | 3.88 | 50.51 | 2.41 | 58.98 | 2.39 | 70.29 |
| | (–2.28) | (+1.11) | (–2.20) | (+0.13) | (–1.84) | (+0.76) | (–0.12) | (–0.16) | (–1.59) | (+0.19) | (–1.61) | (+0.40) |
| | 3.68 | 73.33 | 2.74 | 83.27 | 2.43 | 94.94 | 3.97 | 50.34 | 3.44 | 61.97 | 3.25 | 72.77 |
| | (–0.32) | (+5.55) | (–1.26) | (+4.47) | (–1.57) | (+1.54) | (–0.03) | (–0.33) | (–0.56) | (+3.18) | (–0.75) | (+2.88) |

Table 3: Accuracy (%) and avg. #Calls across five reasoning tasks; Avg is mean over benchmarks. Parentheses show difference vs. SC (#Calls=4 or SC Acc). Efficient (first row) vs. Conservative (second row) are two CGES settings.

| | AIME24 #Calls | AIME24 Acc | MATH500 #Calls | MATH500 Acc | GSM8K #Calls | GSM8K Acc | GPQA #Calls | GPQA Acc | MMLU_Pro #Calls | MMLU_Pro Acc | Avg. #Calls | Avg. Acc |
|---|---|---|---|---|---|---|---|---|---|---|---|---|
| **SC** | 8.00 | 74.45 | 8.00 | 81.47 | 8.00 | 93.91 | 8.00 | 50.17 | 8.00 | 60.68 | 8.00 | 72.14 |
| **CGES (Ours)** | | | | | | | | | | | | |
| **LNS[Arithmetic mean]** | 4.41 | 74.45 | 4.08 | 81.87 | 3.41 | 93.83 | 2.08 | 51.12 | 4.42 | 60.68 | 3.68 | 72.39 |
| | (–3.59) | (0.00) | (–3.92) | (+0.40) | (–4.59) | (–0.08) | (–5.92) | (+0.95) | (–3.58) | (0.00) | (–4.32) | (+0.25) |
| | 6.88 | 74.45 | 4.88 | 81.87 | 4.26 | 93.83 | 7.56 | 51.12 | 6.59 | 60.68 | 6.03 | 72.39 |
| | (–1.12) | (0.00) | (–3.12) | (+0.40) | (–3.74) | (–0.08) | (–0.44) | (+0.95) | (–1.41) | (0.00) | (–1.97) | (+0.25) |
| **LNS[Geometric mean]** | 4.32 | 75.19 | 4.67 | 81.73 | 4.04 | 93.78 | 4.25 | 51.68 | 5.78 | 60.89 | 4.61 | 72.65 |
| | (–3.68) | (+0.74) | (–3.33) | (+0.26) | (–3.96) | (–0.13) | (–3.75) | (+1.51) | (–2.22) | (+0.21) | (–3.39) | (+0.51) |
| | 7.71 | 72.22 | 5.52 | 81.80 | 5.05 | 93.78 | 7.99 | 51.35 | 7.40 | 60.95 | 6.74 | 72.02 |
| | (–0.29) | (–2.23) | (–2.48) | (+0.33) | (–2.95) | (–0.13) | (–0.01) | (+1.18) | (–0.60) | (+0.27) | (–1.26) | (–0.12) |
| **MARS** | 4.04 | 74.45 | 3.69 | 81.60 | 5.08 | 93.88 | 3.56 | 52.69 | 4.43 | 60.75 | 4.16 | 72.67 |
| | (–3.96) | (0.00) | (–4.30) | (+0.13) | (–2.92) | (–0.03) | (–4.44) | (+2.52) | (–3.57) | (+0.07) | (–3.84) | (+0.53) |
| | 7.63 | 75.56 | 5.63 | 81.87 | 5.08 | 93.88 | 7.97 | 51.68 | 7.49 | 61.03 | 6.76 | 72.80 |
| | (–0.37) | (+1.11) | (–2.37) | (+0.40) | (–2.92) | (–0.03) | (–0.03) | (+1.51) | (–0.51) | (+0.35) | (–1.24) | (+0.66) |
| **CGES - Near Ideal Scenario (Ours)** | | | | | | | | | | | | |
| **RM Confidence** | 1.72 | 75.56 | 3.26 | 82.67 | 2.37 | 94.44 | 5.32 | 50.34 | 4.42 | 61.68 | 3.42 | 72.94 |
| | (–6.28) | (+1.11) | (–4.74) | (+1.20) | (–5.63) | (+0.53) | (–2.69) | (+0.17) | (–3.58) | (+1.00) | (–4.58) | (+0.80) |
| | 5.70 | 75.56 | 3.82 | 84.47 | 2.73 | 95.45 | 7.20 | 50.00 | 5.20 | 63.19 | 4.93 | 73.73 |
| | (–2.30) | (+1.11) | (–4.18) | (+3.00) | (–5.27) | (+1.54) | (–0.80) | (–0.17) | (–2.80) | (+2.51) | (–3.07) | (+1.59) |

Table 4: Accuracy (%) and avg. #Calls across five reasoning tasks; Avg is mean over benchmarks. Parentheses show difference vs. SC (#Calls=4 or SC Acc). Efficient (first row) vs. Conservative (second row) are two CGES settings.

Table 6: Comparison of CGES and adaptive self-consistency baselines on 70B–72B models.

| | AIME24 | | MATH500 | | MMLU_Pro | | Avg. | |
|---|---|---|---|---|---|---|---|---|
| | #Calls | Acc | #Calls | Acc | #Calls | Acc | #Calls | Acc |
| **SC** | 16.00 | 83.33 | 16.00 | 87.47 | 16.00 | 74.53 | 16.00 | 81.78 |
| **ESC (w=4)** | 8.98 | 83.33 | 6.30 | **87.60** | 6.67 | 74.54 | 7.32 | 81.83 |
| | (-4.98) | (+0.00) | (-8.12) | (-0.13) | (-10.78) | (-0.02) | (-4.10) | (+0.17) |
| **ESC (w=8)** | 12.62 | 83.33 | 10.34 | 87.47 | 10.88 | 74.52 | 11.28 | 81.77 |
| | (-1.60) | (+0.00) | (-4.43) | (+0.00) | (-6.50) | (+0.00) | (-1.13) | (+0.00) |
| **Adaptive-Consistency (BETA)** | 8.11 | 83.33 | 6.05 | 87.60 | 6.41 | 74.54 | 6.86 | 81.83 |
| | (-5.75) | (+0.00) | (-8.64) | (-0.13) | (-10.95) | (-0.02) | (-4.57) | (+0.50) |
| **CGES (Ours)** | | | | | | | | |
| **LNS[Arithmetic mean]** | **6.86** | 83.33 | **4.47** | 87.20 | **4.37** | 74.54 | **5.24** | 81.69 |
| | (-9.53) | (+0.00) | (-11.31) | (-0.27) | (-11.50) | (-0.13) | (-13.91) | (+0.12) |
| | 8.13 | **84.44** | 4.47 | 87.20 | 5.54 | **74.68** | 6.05 | **82.11** |
| | (-6.41) | (+0.00) | (-10.19) | (-0.33) | (-11.50) | (-0.13) | (-5.52) | (+0.51) |

## C.1 CONFIDENCE CALIBRATION ANALYSIS

To evaluate how sensitive CGES is to miscalibrated confidence estimates, we measured two standard calibration metrics, Expected Calibration Error (ECE) and Maximum Calibration Error (MCE), for all confidence estimators (LNS variants, MARS, and reward-model scores) across all benchmarks. As summarized in Table 5, reward-model confidence is substantially better calibrated on datasets included in its training (e.g., MATH500 and GSM8K), and this corresponds to stronger CGES performance. Conversely, even with noisier estimators such as LNS (Arithmetic), CGES maintains near-identical accuracy and large call reductions, indicating robustness to *imperfect confidence*. *Overall, better calibration primarily improves efficiency (earlier stopping), whereas the final aggregated posterior remains stable, making CGES effective even under significant confidence noise.*

| | AIME24 | | MATH500 | | GSM8K | | GPQA | | MMLU_Pro | |
|---|---|---|---|---|---|---|---|---|---|---|
| | ECE | MCE | ECE | MCE | ECE | MCE | ECE | MCE | ECE | MCE |
| **LNS[Arithmetic mean]** | 0.295 | 0.682 | 0.196 | 0.555 | 0.022 | 0.572 | 0.336 | 0.437 | 0.291 | 0.377 |
| **LNS[Geometric mean]** | 0.195 | 0.610 | 0.153 | 0.375 | 0.036 | 0.451 | 0.205 | 0.359 | 0.189 | 0.302 |
| **MARS** | 0.176 | 0.526 | 0.139 | 0.446 | 0.038 | 0.418 | 0.202 | 0.394 | 0.189 | 0.206 |
| **RM Confidence** | 0.288 | 0.749 | **0.042** | **0.245** | **0.022** | **0.352** | 0.351 | 0.480 | **0.151** | **0.232** |

Table 5: Calibration scores (ECE and MCE) for different confidence estimators. Lower values indicate better calibration.

## C.2 RESULTS WITH 70B–72B MODELS

We additionally evaluated CGES and adaptive SC baselines using large *DeepSeek-R1-Distill-Llama-70B* on AIME24 and *Qwen2.5-72B-Instruct* on MATH500 and MMLU-Pro. As shown in Table 6, the qualitative trends match those observed with 7B models: CGES consistently requires far fewer calls than Self-Consistency and ASC/ESC while achieving comparable or slightly improved accuracy. The Efficient configuration often reduces calls by 60–70% relative to SC, even at 70B scale.

## D  DECODING HYPERPARAMETERS AND PROMPT TEMPLATES

For CGES, we sweep the stopping threshold $\gamma$ over the grid $0.70, 0.75, 0.80, 0.85, 0.90, 0.95, 0.99, 0.999, 0.9999$ to explore different accuracy–efficiency trade-offs. All experiments, including those for SC, ESC, CGES, and confidence estimation, share the same decoding setup. Specifically, we allow a maximum of 32,768 generation tokens, use a temperature of 0.7, and apply nucleus sampling with top-$p = 1.0$ (i.e., no truncation). Top-$k$ sampling is disabled in all runs.

**Prompt Templates.** We use dataset-specific answer-format constraints to simplify parsing. Prompt templates are shown in Fig 4.

Question: {sample['question']}
Provide your step-by-step reasoning first, and then print "The answer is \boxed{X}", where X is the final answer, at the end of your response.

(a) Prompt used for **MATH500, AIME24, GSM8K**

Question: {sample['question']}
Provide your step-by-step reasoning first, and then print "The answer is (X)", where X is the answer choice (one capital letter), at the end of your response.

(b) Prompt used for **GPQA, MMLU_Pro**

Figure 4: Two prompt templates for evaluation.

## E  THE USE OF LARGE LANGUAGE MODELS (LLMS)

We used large language models (LLMs) solely to aid with polishing the writing and improving clarity of exposition. No part of the research ideation, methodology, analysis, or experimental results was generated by LLMs. The authors take full responsibility for the content of this paper.

