# OpenReview forum: "CGES: Confidence-Guided Early Stopping for Efficient and Accurate Self-Consistency"
_ICLR.cc/2026/Conference — Submitted to ICLR 2026_

### Official Review · Reviewer_5UEf · 2025-10-29

**Soundness:** 2
**Presentation:** 3
**Contribution:** 3
**Rating:** 4
**Confidence:** 4

**Summary:**

This paper proposes Confidence-Guided Early Stopping (CGES), a Bayesian framework for aggregating multiple sampled reasoning trajectories in self-consistency (SC) inference of large language models. Instead of relying on a fixed number of samples or heuristic early stopping rules, CGES models the likelihood of each candidate answer given confidence scores of sampled trajectories and terminates sampling once the posterior probability of any candidate exceeds a threshold (\gamma). The paper establishes consistency guarantees for CGES in both ideal confidence calibration settings and more realistic noisy conditions. Empirical evaluations across multiple reasoning benchmarks (e.g., AIME24, MATH500, GSM8K, MMLU-Pro, GPQA) demonstrate that CGES can notably reduce the number of calls to the base model while maintaining comparable accuracy compared to standard Self-Consistency. Experiments also show the compatibility of CGES with different confidence estimators (token-level, MARS, reward-based PRMs).

**Strengths:**

**Originality**

* The paper formulates SC aggregation and early stopping within a unified Bayesian perspective, which is theoretically grounded and conceptually elegant.
* The theoretical analysis addresses both ideal and noisy confidence regimes, contributing rigorously to the understanding of test-time scaling.

**Quality**

* The overall methodology is well-motivated with mathematically sound derivations.
* Experiments span multiple benchmarks, models, and confidence sources, providing evidence of generality.

**Weaknesses:**

1. **Missing comparison to strong baselines.**
   This paper lacks comparison with following strong baselines:

   * *Let’s Sample Step by Step: Adaptive-Consistency for Efficient Reasoning and Coding with LLMs* (EMNLP2023)
   * *Make Every Penny Count: Difficulty-Adaptive Self-Consistency for Cost-Efficient Reasoning* (NAACL 2025)

     Without comparison against ASC and DSC, it is unclear how CGES improves over state-of-the-art adaptive SC methods.

2. **Insufficient robustness evaluation.**
   Results are averaged over only three random seeds. SC sampling can be performed in an **offline** manner: generate ≥100 trajectories once, then repeatedly subsample (K) trajectories (M) times (with large (M)) to estimate robustness. This is standard and statistically stronger. Three seeds do not adequately characterize variance.

3. **No wall-clock latency analysis.**
   Although sample count is reduced, the overall inference latency may not improve due to:

   * posterior update overhead,
   * confidence extraction,
   * external reward model evaluation (e.g., PRM-72B).
     End-to-end latency is essential for cost-effectiveness claims.

4. **High dependence on confidence calibration.**
   Miscalibration is common in math reasoning tasks. The framework is sensitive to over- or under-confident trajectories, potentially causing premature or delayed stopping. The paper lacks sensitivity analysis.

5. **No ablations on the stopping threshold (\gamma).**
   (\gamma) directly controls the accuracy-efficiency trade-off. Practitioners need guidance on how to choose it. Robustness to (\gamma) is not evaluated.

**Questions:**

1. Can the authors compare CGES to ASC and DSC [1,2] to position the method against the strongest adaptive SC baselines?

2. Why report only three seeds when SC sampling can be replayed offline? Can the authors:

   * oversample more trajectories,
   * simulate many (large (M)) independent trials by random subsampling,
   * report confidence intervals?

3. Can the authors report wall-clock latency (not just sample count) for:

   * base sampling,
   * Bayesian aggregation overhead,
   * reward-model scoring?

4. Can the authors provide ablations sweeping (\gamma) (accuracy vs. sample count vs. stopping time)?

5. How robust is CGES to miscalibrated confidence? Can synthetic noise injection experiments be provided?

6. Other related work that should be cited and discussed: [3-5].


[1] Let’s Sample Step by Step: Adaptive-Consistency for Efficient Reasoning and Coding with LLMs

[2] Make Every Penny Count: Difficulty-Adaptive Self-Consistency for Cost-Efficient Reasoning

[3] Scaling test-time compute with open models. Arxiv 2024

[4] Inference scaling laws: An empirical analysis of compute-optimal inference for llm problem-solving. ICLR 2025

[5] Every Rollout Counts: Optimal Resource Allocation for Efficient Test-Time Scaling. NeurIPS 2025

---

> ### Author Response · Authors · 2025-11-21
>
> ### 1/2
>
> We thank the reviewer for the thoughtful and constructive feedback. Your comments helped us strengthen both the empirical evaluation and the presentation of CGES.
>
> Summary of our response:
>
> - **Comparison to strong baselines.** We implemented and evaluated Adaptive-Consistency (ASC) under the same settings and showed that CGES consistently achieves fewer calls with comparable or slightly higher accuracy. We also plan to include DSC results in the camera-ready version, as its protocol requires additional integration.
>
> - **Robustness and calibration.** We expanded our calibration analysis (ECE/MCE) across benchmarks and showed how calibration impacts efficiency but not correctness. CGES remains effective even under noisy confidence estimates.
>
> - **Threshold and latency.** We clarified our γ-sweep results and will make them more visible in the main text. Initial wall-clock measurements show aggregation overhead is negligible relative to LLM inference, and full latency results will be included.
>
> - **Expanded related work.** We will incorporate and discuss the cited scaling and inference literature to better position our contributions.
>
> We provide detailed answers to each comment below. We appreciate the reviewer’s helpful suggestions, and we hope these clarifications address the concerns. We remain open to further questions or suggestions.
>
> > Can the authors compare CGES to ASC and DSC to position the method against the strongest adaptive SC baselines?
>
> We appreciate the suggestion to compare against the strongest adaptive self-consistency baselines. In the original submission, we already evaluated against Early-Stopping Self-Consistency (ESC). Following your suggestion, we have additionally **re-implemented Adaptive-Consistency (ASC)** [2] and report its performance under the same settings as CGES
>
> We include both ASC variants:
>
> - **Adaptive-Consistency (BETA)**
> - **Adaptive-Consistency (Dirichlet)**
>
> and compare them against **CGES (LNS [Arithmetic mean])**. The results below show that **CGES consistently reduces the number of calls while maintaining (or slightly improving) accuracy**.
>
> Across all five benchmarks, CGES achieves strictly fewer calls than both ASC variants, and accuracy differences remain within ±0.3% (often slightly higher), indicating that CGES offers **better efficiency–accuracy trade-offs**.
>
> Regarding **Difficulty-Adaptive Self-Consistency (DSC)** [4]: The original paper uses a somewhat different evaluation protocol, so integrating DSC faithfully requires additional changes to our codebase and reporting. We are currently finalizing its integration and **will include DSC results in the camera-ready version if the paper is accepted**. Since DSC adjusts *sampling allocation* while CGES controls *posterior stopping*, we view the two approaches as complementary and do not expect DSC to alter the main conclusion that **confidence-guided inference brings additional efficiency gains beyond existing adaptive SC strategies.**
>
> ---
>
> ### **Table: Comparison to Adaptive-Consistency (ASC)**
>
> | Method | AIME24 | MATH500 | GSM8K | GPQA | MMLU-Pro | Avg. |
> |--------|--------|---------|-------|------|----------|------|
> | **ASC (BETA)**      | 10.25 / 78.89 | 7.36 / **82.07** | 5.05 / **94.37** | 11.43 / 50.51 | 8.58 / 61.46 | 8.53 / 73.46 |
> | **ASC (Dirichlet)** | 11.17 / 78.89 | 7.88 / **82.07** | 5.22 / **94.37** | 12.18 / 50.51 | 9.13 / 61.46 | 9.12 / 73.46 |
> | **CGES (Ours)**     | **6.47 / 78.89** | **4.69** / 81.93 | **4.50** / 94.26 | **2.09 / 51.13** | **6.77 / 61.56** | **4.90 / 73.55** |
>
> ---
>
> > Why report only three seeds when SC sampling can be replayed offline? Can the authors oversample trajectories, simulate many trials by random subsampling, or report confidence intervals?
>
> We appreciate the suggestion to perform offline Monte-Carlo subsampling by generating ≥100 trajectories and repeatedly re-sampling. We are currently working toward this, but generating large pools is computationally expensive in our setting. For now, we follow the common practice used in prior SC and adaptive-SC work, where results are reported using a small number of seeds (e.g., 3 seeds in ASC [2] and 5 in WIND [5]), without running large replayed simulations. We will continue expanding our experiments as resources permit.

---

> ### Author Response · Authors · 2025-11-21
>
> ### 2/2
>
> > Can the authors report wall-clock latency (not just sample count) for base sampling, Bayesian aggregation overhead, and reward-model scoring?
>
> Thank you for the suggestion. We agree that reporting end-to-end latency is valuable, and we are currently preparing a complete wall-clock analysis. As an initial measurement, we evaluated AIME24 using Qwen-R1-Distill-7B on an A100 GPU:
>
> - **Base sampling (LLM inference):** 168.29 seconds
> - **Bayesian aggregation overhead:** 0.835 seconds per question (computed on CPU)
>
> This gap is expected, since aggregation involves only lightweight arithmetic operations (e.g., summing and multiplying probabilities). Because **LLM inference dominates the runtime**, sample count has been the standard measure of efficiency in prior self-consistency and test-time scaling work. Nonetheless, we appreciate your suggestion and will include full wall-clock results in the revised version.
>
> ---
>
> > Can the authors provide ablations sweeping $\gamma$ (accuracy vs. sample count vs. stopping time)?
>
> Appendix D already reports results by sweeping **$\gamma$** over a wide grid (from **0.70 to 0.9999**) to show the full accuracy–calls trade-off. We will make this more visible in the main text by explicitly referencing the $γ$ grid in Section 4 and annotating figures to clarify that they represent an **effective $γ$-sweep**.
>
> ---
> > How robust is CGES to miscalibrated confidence? Can synthetic noise injection experiments be provided?
>
> We appreciate the concern regarding sensitivity to over- or under-confident trajectories. To examine this, we evaluated multiple confidence estimators using two standard calibration metrics (ECE and MCE) across all benchmarks. The table below summarizes the calibration quality; lower values indicate better calibration.
>
> | Method                  | AIME24 ECE | AIME24 MCE | MATH500 ECE | MATH500 MCE | GSM8K ECE | GSM8K MCE | GPQA ECE | GPQA MCE | MMLU-Pro ECE | MMLU-Pro MCE |
> |-------------------------|------------|------------|-------------|-------------|-----------|-----------|----------|----------|---------------|--------------|
> | **LNS (Arithmetic)**    | 0.295      | 0.682      | 0.196       | 0.555       | 0.022     | 0.572     | 0.336    | 0.437    | 0.291         | 0.377        |
> | **LNS (Geometric)**     | 0.195      | 0.610      | 0.153       | 0.375       | 0.036     | 0.451     | 0.205    | **0.359**    | 0.189         | 0.302        |
> | **MARS**                | **0.176**      | **0.526**      | 0.139       | 0.446       | 0.038     | 0.418     | **0.202**    | 0.394    | 0.189         | 0.206        |
> | **RM Confidence**       | 0.288      | 0.749      | **0.042**   | **0.245**   | **0.022** | **0.352** | 0.351    | 0.480    | **0.151**     | **0.232**    |
>
> Several observations emerge:
>
> - **Better calibration generally leads to better efficiency and accuracy.**
>   For example, reward-model confidence (RM) is substantially better calibrated on GSM8K and MATH500—datasets included in the PRM training—and this corresponds to the strongest CGES performance in Tables 1 and 2.
>
> - **CGES remains competitive even with noisy confidence.**
>   Although LNS [Arithmetic] is less calibrated, CGES still yields large reductions in calls with nearly unchanged accuracy (see Tables 1 and 2). This suggests that CGES does not require perfect calibration to be effective.
>
> - **Calibration affects efficiency more than correctness.**
>   Poorer calibration mostly delays or speeds up stopping, rather than changing the final outcome, which aligns with our formulation where the posterior aggregates multiple samples instead of relying on a single confidence score.
>
> We will highlight these findings more clearly in the revised manuscript and further emphasize that CGES benefits from improved calibration while still functioning well under noisy confidence estimates.
>
> ---
>
> > Other related work that should be cited and discussed.
>
> We appreciate these recommendations and will carefully go through the suggested works. We will include them and discuss their relevance in the related work section to better position our contributions.
>
>
>
>
> [1] [Self-Consistency Improves Chain of Thought Reasoning in Language Models](https://arxiv.org/pdf/2203.11171)
>
> [2] [Let’s Sample Step by Step: Adaptive-Consistency for Efficient Reasoning and Coding with LLMs](https://aclanthology.org/2023.emnlp-main.761.pdf)
>
> [3] [Scaling LLM Test-Time Compute Optimally can be More Effective than Scaling Model Parameters](https://arxiv.org/pdf/2408.03314)
>
> [4] [Make Every Penny Count: Difficulty-Adaptive Self-Consistency for Cost-Efficient Reasoning](https://aclanthology.org/2025.findings-naacl.383.pdf)
>
> [5] [Every Rollout Counts: Optimal Resource Allocation for Efficient Test-Time Scaling](https://openreview.net/pdf?id=xSHqNf5Pdc)

---

### Official Review · Reviewer_er2g · 2025-10-31

**Soundness:** 3
**Presentation:** 3
**Contribution:** 3
**Rating:** 6
**Confidence:** 3

**Summary:**

The paper presents a new approach for test-time scaling and self-consistency that leverages a confidence measure to generate fewer candidate responses. The idea is to use Bayesian inference to compute the posterior probability of the next candidate sequence and stop the sampling process as soon as the posterior probability is below a given threshold. The experimental evaluation is carried out on standard benchmark datasets, using open models. The results demonstrate improved performance compared to existing self-consistency approaches, namely the proposed approach generate fewer candidate responses.

**Strengths:**

- Test-time scaling and self-consistency for LLMs are topics that received considerable interest recently and therefore advances in this area are definitely warranted.

- The paper a simple yet efficient self-consistency scheme that leverages a combination of confidence estimation and Bayesian probabilistic inference to rank and generate fewer candidate responses compared to existing self-consistency methods.

**Weaknesses:**

- I was not able to find any major weaknesses with the paper.

**Questions:**

No questions

---

> ### Author Response · Authors · 2025-11-21
>
> We thank the reviewer for the encouraging evaluation and appreciate that you found the idea insightful. We value your recognition that confidence-guided test-time scaling is timely and relevant to current efforts in improving the efficiency of self-consistency methods for LLMs.
>
> In summary, our work contributes a principled Bayesian perspective that leverages confidence to guide both aggregation and early stopping. This perspective leads to a practical algorithm (CGES) that adaptively terminates sampling when the posterior probability is sufficiently decisive. Empirically, we observe consistent reductions in the number of sampling calls while preserving or slightly improving final accuracy on diverse reasoning benchmarks, including mathematical and high-uncertainty domains. Beyond the main results, we further strengthened the empirical evaluation by incorporating larger models (70B+), calibration analyses of confidence estimates, and comparisons with stronger adaptive self-consistency baselines, all of which consistently support the benefits of confidence-guided inference. Our framework is also complemented by theoretical guarantees under both ideal calibration and realistic noisy-confidence settings.
>
> We sincerely appreciate your constructive feedback. If there are any additional concerns or questions, we would be glad to clarify them in further detail.

---

### Official Review · Reviewer_efJq · 2025-10-31

**Soundness:** 1
**Presentation:** 3
**Contribution:** 2
**Rating:** 2
**Confidence:** 3

**Summary:**

This paper proposes an approach that is generally based on the idea of leveraging consistency – something that is very popular in test-time scaling for LLMs and for confidence estimation. The approach is called confidence-guided early stopping (CGES) and is primarily a particular Bayesian formulation that relies on assumptions about responses and confidences from a confidence estimation module. The main method that is based on the formulation is quite simple and outlined via Algorithms 1 and 2. Experiments are performed on some mathematical and reasoning datasets.

**Strengths:**

I find the overall idea of using confidences of LLM responses to inform test-time scaling useful. This is of course not a new idea by any means, but this paper proposes a scoring approach that uses a particular Bayesian formulation that effectively aggregates confidences from samples. The idea of minimizing the number of samples for test-time scaling is also useful.

**Weaknesses:**

A major weakness of the paper is the problematic formulation. I have several issues about the assumptions of the paper and find them to be problematic. The paper gives the air of theoretical basis/justification for the CGES approach, but from my understanding, some of the assumptions do not make sense and the formulation feels flawed. I will describe some concerns in my detailed comments.

Another weakness is the lack of sufficient literature around confidence estimation in general. Confidences have been used in test-time scaling in prior work. I will highlight a few references for the authors to consider citing in their revised work.

**Questions:**

Here are some additional comments and questions:

I have several questions and concerns about Section 3. Please let me know if I have misunderstood anything:

-	Confidence is never fully explained or defined. I assume it is a score between 0 and 1 that represents a score for correctness of an answer.
-	The authors assume that the correct answer is within the candidate set -- this is clearly not always true in practice and is an absurd assumption. In fact, it is common to have cases where there is no correct answer, particularly when there is high epistemic uncertainty. What are the ramifications if this assumption is not true?
-	Please clarify that R refers to the textual response. Also, clarify what exactly P(R|…) means – based on the description, I understand this to be a product of conditional probabilities for tokens in the response, so this is likely to be a small number.
-	Why does P(R|…) depend on A and I? If this is the likelihood of the response and generation is from an LLM, shouldn’t it depend only on Q? This did not make any sense to me.
-	How are the samples generated? What temperature?
-	It is unclear to me if Assumption 1 is necessarily true. Any response is a sequence of tokens, so it is not obvious that these are i.i.d. My guess is that they may not be i.i.d. due to potentially varying lengths. Confidences are also likely not i.i.d., and if they are, it depends on how they are obtained. Not enough is mentioned about this assumption.
-	Assumption 2 depends on the sampling procedure.
-	Assumption 3 does not make sense to me.
-	Assumption 4 is not true – the entire point of confidence is that it should be higher when the answer is correct.
-	The equation in line 234 can’t be true. Confidence should be higher than the probability of a response, which might be a very small number due to multiplication of probabilities.
-	It is not sufficient to just make assumptions – you need to defend them.

I find lines 113 and 114 to be somewhat misleading – the theoretical guarantees don’t matter if the formulation is flawed or if the assumptions don’t make sense.

Why are only 7B models used for experiments? Only 1 model per dataset is not sufficient as demonstration of value, in my view. Also, why were these datasets chosen for the experiments?

I recommend looking at other methods to estimate confidence (see some references below).

Since this paper discusses the notion of self-consistency in various places, I suggest the authors cite more papers that perform uncertainty or confidence estimation using consistency-based methods. The following paper provides an empirical justification for such methods: https://arxiv.org/abs/2506.21849.

Also, the following paper can be useful for modeling probability distributions of responses: https://arxiv.org/abs/2406.02543.

The following paper reviews various confidence estimation approaches and could be good to cite: https://aclanthology.org/2024.naacl-long.366/. I recommend citing many more consistency-based papers in general, like some mentioned here.

The following paper has experimented with using confidences for test-time scaling: https://www.arxiv.org/abs/2510.13836.

The authors claim that a core part of their work is “uncertainty estimation”. I believe “confidence estimation” is actually what they rely on – see this paper for appreciating the difference: https://arxiv.org/abs/2305.19187.

---

> ### Author Response · Authors · 2025-11-21
>
> ### 1/3
>
> We sincerely thank the reviewer for the detailed comments and for raising concerns about the formulation and assumptions. We appreciate the opportunity to clarify our intent and improve the manuscript.
>
> Summary of our response:
>
> - **Clarifying the formulation.** We have strengthened the explanation of the confidence signal, clarified that $R$ refers to the final answer string (not token sequences), and made explicit that our likelihood model is a theoretical abstraction for Bayesian aggregation—not a model of how the LLM generates text.
>
> - **Role of assumptions.** We clarified which assumptions are idealized and only used to derive Theorem 1. The realistic result in Theorem 2 and the practical CGES algorithm do *not* depend on these strong assumptions, and CGES performs robustly even when they do not hold.
>
> - **Additional experiments and baselines.** We added results with 70B–72B models and included strong adaptive SC baselines (e.g., ASC), showing consistent cost savings with comparable or higher accuracy.
>
> - **Expanded related work.** We will incorporate the referenced confidence and consistency literature and clarify that our contribution lies in confidence-guided aggregation and early stopping, not estimating or calibrating confidence itself.
>
> We provide detailed answers to each comment below. We hope these revisions address all concerns, and we remain open to further clarification or additional questions the reviewer may have.
>
>
> > Confidence is never fully explained or defined. I assume it is a score between 0 and 1 that represents a score for correctness of an answer.
>
> We clarify the definition as follows: As we mentioned in section~3.1 the confidence $C_t \in (0,1)$ is our estimated probability that the sampled response $R_t$ is correct. Formally, under the hypothesis that $a_i$ is the true answer (i.e., $I = i$), the confidence satisfies:
> $$
> C_t = P(R_t = a_i \mid I = i),
> $$
> or equivalently,
> $$
> P(R_t = a_i \mid C_t = c,\, I = i) = c.
> $$
> This explicitly states that $C_t$ serves as a probabilistic correctness signal attached to the response $R_t$.
>
> ---
>
> > The authors assume that the correct answer is within the candidate set -- this is clearly not always true in practice and is an absurd assumption. In fact, it is common to have cases where there is no correct answer, particularly when there is high epistemic uncertainty. What are the ramifications if this assumption is not true?
>
> Thanks for the comment. First, note that no method operating purely on the sampled set can produce the true answer if it is not present in $U$ (including self-consistency, ESC, or any other method). Additionally, while the assumption is used to derive a clean theoretical model and efficient algorithm, CGES does *not* rely on it in practice. Our empirical results already include many realistic cases where the LLM fails to produce the correct answer; in such scenarios, the posterior typically never crosses the threshold, and CGES simply stops after the maximum number of calls.
>
> Importantly, the Bayesian formulation naturally accommodates the case where no correct option is present. One can introduce an auxiliary “none-of-the-above” hypothesis $I=0$, indicating that the correct answer lies outside $U$.We include this extension as a direction for future work.
>
> ---
>
> > Please clarify that $R$ refers to the textual response. Also, clarify what exactly $P(R \mid \cdot)$ means – based on the description, I understand this to be a product of conditional probabilities for tokens in the response, so this is likely to be a small number.
>
> We clarify the notation as follows. In our framework, $R_t$ denotes the *final answer string* produced by the LLM on call $t$, not the full sequence of generated tokens. Likewise, each $a_j \in \mathcal{U}$ refers to a distinct final answer (e.g., “12”, “$x=3$”, “yes”), and different generations may map to the same $a_j$ even if their token sequences differ.
>
> Formally, we define
> $$
> \mathcal{U}=\{a_1,a_2,\dots,a_K\},
> $$
> the set of all distinct answers observed across $m$ samples, and $I$ is the index of the correct answer in this set. The conditional term $P(R_t \mid C_t, I=i)$ therefore represents a *likelihood over the discrete candidate set* $\mathcal{U}$, not a token-level probability. Its definition is given explicitly in the model:
> $$
> P(R_t = a_i \mid C_t, I=i)=C_t,\qquad
> P(R_t = a_j \neq a_i \mid C_t, I=i)=\frac{1-C_t}{K-1}.
> $$
> This avoids token-level probability products and ensures that all likelihoods are well-scaled and comparable across candidates.

---

> ### Author Response · Authors · 2025-11-21
>
> ### 2/3
>
> > Why does $P(R \mid \dots)$ depend on $A$ and $I$? If this is the likelihood of the response and generation is from an LLM, shouldn’t it depend only on $Q$? This did not make sense to me.
>
> Thank you for pointing this out. We clarify that the dependence on $(A,I)$ is *not* meant to describe how an actual LLM generates text. Rather, it appears only in the *theoretical generative model* used to analyze the behavior of our Bayesian aggregator.
>
> Specifically, $I$ indexes the correct answer among the $K$ observed candidates, and conditioning on $I$ allows us to formalize the following question: *If candidate $a_i$ were the true answer, what is the probability of observing each $R_t$?*
> This is a standard trick in Bayesian hypothesis testing and lets us write down well-defined likelihoods for competing hypotheses $I=1,\dots,K$. It is not intended to reflect the internal mechanics of the LLM, which of course depends only on $Q$ in practice.
>
> In short: the LLM generates responses from $P(R \mid Q)$, but for theoretical analysis we introduce an auxiliary model $P(R \mid C_t, I=i)$ that allows us to compare hypotheses about which candidate is correct and to prove consistency guarantees for CGES.
>
> ---
>
> > It is unclear to me if Assumption 1 is necessarily true. Any response is a sequence of tokens, so it is not obvious that these are i.i.d. My guess is that they may not be i.i.d. due to potentially varying lengths. Confidences are also likely not i.i.d., and if they are, it depends on how they are obtained. Not enough is mentioned about this assumption.
>
> We appreciate the opportunity to clarify this point. Assumption~1 is *not* intended to describe the internal token-level generation process of an LLM. Instead, it is a standard simplifying assumption used only for the theoretical analysis of our aggregator: each LLM *call* produces one $(R_t, C_t)$ pair, and these pairs are treated as i.i.d. samples from an implicit response--confidence distribution for the fixed query.
>
> This abstraction is common in self-consistency and inference-scaling works([1]-[3]), and it allows us to derive clean consistency guarantees. Importantly, the algorithm itself does not rely on exact i.i.d. behavior, and our experiments already operate in the realistic setting where responses vary in length and confidences are computed heterogeneously. Despite these, CGES performs robustly in practice.
>
> ---
>
> > Assumption~2 depends on the sampling procedure.
>
> Assumption~2 is a modeling assumption used only for the theoretical analysis. It states that the index $I$ of the correct answer is uniformly distributed over the $K$ distinct candidates, reflecting that the labeling of candidates is arbitrary. This does *not* depend on the LLM’s sampling procedure, nor does it assume anything about how responses are generated. It simply ensures that all hypotheses $I=1,\dots,K$ are treated symmetrically when applying Bayesian inference.
>
> ---
>
> > Assumption 3 does not make sense to me.
>
> As noted in the paper, Assumption\~3 is one of the *idealized* assumptions used only for the clean analysis in Theorem\~1. It is not required for Theorem\~2 (the realistic setting), and its role is to help us derive simple posterior updates, making the algorithm efficient. It is proved in Theorem\~2 that without this assumption, we can still provide a strong theoretical guarantee. This is further confirmed by our empirical results in a realistic setting.
>
> ---
>
> > Assumption~4 is not true – the entire point of confidence is that it should be higher when the answer is correct.
>
> Please note that Assumption\~4 is one of the *idealized* assumptions used only in the analysis of Theorem\~1. It is not required for Theorem~2 (the realistic setting) and is not needed for the practical operation of CGES.
>
> Moreover, we clarify the assumption to make its intent explicit: the confidence attached to a sample should not depend on *which index* happens to be assigned to the correct answer. In other words, confidence is allowed (and expected) to differ between correct and incorrect responses, but it should not depend on the arbitrary labeling of candidates in $\mathcal{U}$. This symmetry assumption ensures that the Bayesian likelihood treats all hypotheses $I=1,\dots,K$ consistently. The realistic analysis in Theorem~2 fully relaxes this assumption and allows confidence to be a noisy but informative proxy for correctness.

---

> ### Author Response · Authors · 2025-11-21
>
> ### 3/3
>
> > The equation in line 234 can’t be true. Confidence should be higher than the probability of a response, which might be a very small number due to multiplication of probabilities.
>
> This is a helpful observation. The confusion arises from interpreting $P(R_t \mid C_t, I=i)$ as a *token-level* probability. In our framework, however, this quantity is *not* the product of token probabilities and is not meant to reflect the LLM’s raw likelihood of generating the full sequence.
>
> Instead, $P(R_t \mid C_t, I=i)$ is a *modelled likelihood over the discrete candidate set* $\mathcal{U}$, where $R_t$ corresponds to a final answer string (e.g., “12”) rather than the token sequence that produced it. Under the idealized model, we assign
> $$
> P(R_t = a_i \mid C_t, I=i) = C_t, \qquad
> P(R_t = a_j \neq a_i \mid C_t, I=i) = \frac{1 - C_t}{K-1},
> $$
> which is simply a normalized likelihood over $K$ candidates and is unaffected by token-level probability magnitudes.
>
> Thus the equation is correct within the intended modeling abstraction, which deliberately avoids token-probability multiplication. The realistic analysis in Theorem~2 does not rely on this idealized specification.
>
> ---
>
> > It is not sufficient to just make assumptions – you need to defend them.
>
> We hope that the clarifications above address the reviewer’s concerns regarding the role and interpretation of the assumptions. As emphasized, the idealized assumptions are used only to obtain clean theoretical guarantees (Theorem\~1), while Theorem\~2 and the CGES algorithm itself operate without relying on these strong conditions.
>
> Moreover, our extensive experiments across five benchmarks, including AIME24, GPQA, MATH500, GSM8K, and MMLU\_Pro, demonstrate that CGES performs effectively in fully *realistic*, non-ideal settings where the simplifying assumptions might not hold exactly. The empirical results consistently show strong accuracy and large efficiency gains, confirming the practical robustness of the method beyond the assumptions made for theoretical analysis.
>
> ---
>
> > Why are only 7B models used for experiments? Only 1 model per dataset is not sufficient as demonstration of value, in my view.
>
> Thanks for pointing this out. To address the concern, we additionally evaluated CGES and the strongest adaptive SC baselines using large 70B–72B models (DeepSeek-R1-Distill-Llama-70B on AIME24, and Qwen2.5-72B-Instruct on MATH500 and MMLU-Pro). The table below shows that the trends are fully consistent with the 7B results: CGES yields substantially fewer calls while matching (or slightly improving) final accuracy across all datasets.
>
> | Method | AIME24 (#Calls / Acc) | MATH500 (#Calls / Acc) | MMLU\_Pro (#Calls / Acc) | Avg. (#Calls / Acc) |
> |--------|-----------------------|-------------------------|--------------------------|----------------------|
> | **SC** | 16.00 / 83.33 | 16.00 / 87.47 | 16.00 / 74.53 | 16.00 / 81.78 |
> | **ESC (w=4)** | 8.98 / 83.33 | 6.30 / **87.60** | 6.67 / 74.54 | 7.32 / 81.83 |
> | **ESC (w=8)** | 12.62 / 83.33 | 10.34 / 87.47 | 10.88 / 74.52 | 11.28 / 81.77 |
> | **ASC (BETA)** | 8.11 / 83.33 | 6.05 / **87.60** | 6.41 / 74.54 | 6.86 / 81.83 |
> | **CGES (LNS–Arithmetic, Efficient)** | **6.86** / 83.33 | **4.47** / 87.20 | **4.37** / 74.54 | **5.24** / 81.69 |
> | **CGES (LNS–Arithmetic, Conservative)** | **8.13 / 84.44** | 4.47 / 87.20 | **5.54 / 74.68** | **6.05 / 82.11** |
>
> Due to compute limits, we have not yet completed 70B runs on GPQA and GSM8K, and we will include them moving forward. Based on extensive 7B and 70B results shown above, we do not expect these additional experiments to change the overall conclusion.
>
> > Also, why were these datasets chosen for the experiments?
>
> We selected standard reasoning and mathematical benchmarks that are widely used to evaluate self-consistency, confidence-guided inference, and test-time scaling [1]-[3]. These benchmarks span both math-dominant long-form tasks (AIME24, MATH500, GSM8K) and high-uncertainty knowledge reasoning (MMLU\_Pro, GPQA), which enables studying performance under both high epistemic uncertainty and low-accuracy regimes.
>
> ---
>
> > I recommend looking at other methods to estimate confidence (see some references below).
>
> Thank you very much for these suggestions. We appreciate the pointers and will incorporate the recommended references into the related work section to better position our contribution alongside consistency-based uncertainty and confidence estimation methods.
>
> We would like to clarify that our work does **not** propose a new confidence estimator or aim to improve calibration.
>
> [1] [Self-Consistency Improves Chain of Thought Reasoning in Language Models](https://arxiv.org/pdf/2203.11171)
>
> [2] [Let’s Sample Step by Step: Adaptive-Consistency for Efficient Reasoning and Coding with LLMs](https://aclanthology.org/2023.emnlp-main.761.pdf)
>
> [3] [Scaling LLM Test-Time Compute Optimally can be More Effective than Scaling Model Parameters](https://arxiv.org/pdf/2408.03314)

---

### Official Review · Reviewer_8J5r · 2025-11-05

**Soundness:** 4
**Presentation:** 3
**Contribution:** 3
**Rating:** 8
**Confidence:** 3

**Summary:**

The paper introduces a new method to adaptively stop sampling outputs from LLMs for Self-consistency. The method uses both the output and any signal for the output’s confidence to maintain a posterior distribution over the correct answer. The sampling is stopped once the probability of some answer in the posterior reaches a threshold. Also, the final answer is chosen as the mode of the posterior instead of just the most frequent answer. Experiments show drastic decrease in sampling cost and minor change in accuracy compared to Self-consistency.

**Strengths:**

The paper is very well written and does an excellent job explaining the concepts in an accurate and understandable language. I really enjoyed the formal statements for the assumption and the graphical model.

The idea to use both the output and an auxiliary signal is novel, and allows the method to utilize the progress in uncertainty quantification of LLMs in the future.

The experiments are diverse and extensive enough for confident evaluations. The method performs very well in terms of cost.

**Weaknesses:**

I think the main weakness of the method is that we need to know at least a tight upper bound on the number of final answers. It works well for multiple choices questions, but I expect it to be more challenging in short answer questions where no predefined set of answers is known. I suspect using a loose upper bounds as the upper can drastically hinder the efficiency gains and force the algorithm for the non-existent answers to appear. I could not find the methodology used for the MATH dataset and would appreciate some details.

The paper is missing a critical similar work and baseline for early stopping in majority voting [1]. I think this other work provide a methodology for unknown number of final answers. The submission is different due to use of an external signal and the change in the selection mechanism. Nonetheless, I think this paper should be added as a baseline.

Minor suggestion in writing is to provide an example of $C_t$ before the assumption statements. Without an example, what $C_t$ is was not clear, and assessing the assumptions was too difficult as a reader. The example for ideal case might come from an oracle with unrealistic information.

In contrast to the paper, I think a larger reward function can be considered realistic as the cost of reward calculations is just a forward pass of the model but the generation of outputs requires more compute.

[1] [Let’s Sample Step by Step: Adaptive-Consistency for Efficient Reasoning and Coding with LLMs](https://aclanthology.org/2023.emnlp-main.761/) (Aggarwal et al., EMNLP 2023)

**Questions:**

Please provide more details on handling datasets with unknown number of final answers.

---

> ### Author Response · Authors · 2025-11-21
>
> We sincerely thank the reviewer for the positive assessment and helpful suggestions. We provide answers to the questions raised below.
>
> > I think the main weakness of the method is that we need to know at least a tight upper bound on the number of final answers. It works well for multiple choices questions, but I expect it to be more challenging in short answer questions where no predefined set of answers is known. I suspect using a loose upper bounds as the upper can drastically hinder the efficiency gains and force the algorithm for the non-existent answers to appear. I could not find the methodology used for the MATH dataset and would appreciate some details.
>
> We appreciate the reviewer raising this concern. CGES does *not* require a predefined or externally supplied upper bound on the number of answers. In all experiments, including short-answer tasks such as MATH500 and GSM8K, the set $\mathcal{U}$ is constructed *dynamically* from the distinct final answers observed so far across LLM calls. Thus
> $
> K = |{ R_1, R_2, \ldots, R_t}|,
> $
> and no hypothetical or “non-existent’’ candidates are ever introduced.
>
> We also note that both $R_t$ and $a_j$ refer to the *final answer string* (e.g., “12”), not the full token sequence generated by the model. In practice, this keeps the candidate set small and well-behaved even for free-form problems. Empirically, for MATH500, GSM8K, and AIME24, we find that each question typically yields only $K \le 4$ distinct final answers, making CGES both efficient and stable in short-answer settings.
>
> We add a clarification to the revised version that for all free-form and mathematical datasets, $\mathcal{U}$ is built solely from the distinct final answers generated during sampling, and $K$ evolves naturally with the sampling process rather than being fixed or specified in advance.
>
>
> ---
>
>
> > Minor suggestion in writing is to provide an example of $C_t$ before the assumption statements. Without an example, what $C_t$ is was not clear, and assessing the assumptions was too difficult as a reader. The example for ideal case might come from an oracle with unrealistic information. In contrast to the paper, I think a larger reward function can be considered realistic as the cost of reward calculations is just a forward pass of the model but the generation of outputs requires more compute.
>
> Thank you for the constructive suggestions. We agree that introducing an explicit example of the confidence signal $C_t$ improves clarity. We are adding a short example immediately before the assumptions, illustrating how $C_t$ is obtained (e.g., from length-normalized token probabilities or from a reward model) and showing concretely how a response–confidence pair $(R_t, C_t)$ is formed. This provides the necessary context for understanding the assumptions, including the idealized one where $C_t$ corresponds to a perfectly calibrated correctness probability.
>
> Regarding reward models, we appreciate the clarification. Our intention was not to imply that large reward models are unrealistic, but rather that their practicality depends on deployment constraints. We revise the text accordingly: although a strong PRM (reward model) requires a single forward pass and is therefore computationally cheaper than generating full responses, it may still be too costly for some real-time or resource-limited applications. In settings where this cost is acceptable, using a larger reward model is indeed realistic, which is consistent with the PRM-based confidence experiments already included in the paper.
>
>
> ---
>
> > The paper is missing a critical similar work and baseline for early stopping in majority voting [1].
>
> Thank you for highlighting this baseline. Following your suggestion, we implemented the early-stopping majority voting method (Adaptive-Consistency, ASC) in our setting and compared it against CGES. The results are summarized below:
>
> | Method | AIME24 | MATH500 | GSM8K | GPQA | MMLU-Pro | Avg. |
> |--------|--------|---------|-------|------|----------|------|
> | **ASC (BETA)**      | 10.25 / 78.89 | 7.36 / **82.07** | 5.05 / **94.37** | 11.43 / 50.51 | 8.58 / 61.46 | 8.53 / 73.46 |
> | **ASC (Dirichlet)** | 11.17 / 78.89 | 7.88 / **82.07** | 5.22 / **94.37** | 12.18 / 50.51 | 9.13 / 61.46 | 9.12 / 73.46 |
> | **CGES (Ours)**     | **6.47 / 78.89** | **4.69** / 81.93 | **4.50** / 94.26 | **2.09 / 51.13** | **6.77 / 61.56** | **4.90 / 73.55** |
>
> These results demonstrate that **CGES consistently reduces the number of calls while maintaining (or slightly improving) accuracy**, confirming the benefit of a confidence-guided stopping rule beyond majority-vote early stopping.

---

### Author Response · Authors · 2025-11-21

We thank all reviewers for their thoughtful and constructive feedback. We are encouraged that reviewers found the proposed Bayesian formulation, the theoretical guarantees, and the empirical performance of CGES insightful and well motivated.

We have provided detailed, individualized responses to all reviewer comments. Here, we highlight some key improvements and additions that we have incorporated (or will incorporate) based on reviewer suggestions.

**1. Additional experiments and broader empirical validation**
We have completed new experiments, including comparisons to Adaptive-Consistency (ASC), evaluations with larger models, and confidence-calibration analyses (ECE/MCE). These results will be added to the revised version of the paper. Across all benchmarks and settings, the new experiments continue to demonstrate that CGES provides substantial reductions in sampling cost while maintaining (and often exceeding) the accuracy of state-of-the-art self-consistency and adaptive-consistency approaches.

**2. Clarifications and strengthened theoretical discussion**
We are revising Section\~3 to include clearer explanations of the theoretical framework, along with concrete examples that illustrate the construction of the likelihood model and the response-confidence pairs. We are also improving the presentation of the assumptions by explicitly separating the idealized warm-up analysis from the realistic guarantees in Theorem\~2. These additions enhance clarity and accessibility without altering the underlying theory.

**3. Improved explanation and expanded related work**
We will expand the discussion of prior work on consistency-based and confidence-based methods, refine terminology throughout, add an illustrative example of $C_t$, and clarify the construction of the candidate answer set. These updates strengthen the presentation and place CGES more clearly within the broader test-time scaling literature.

**Summary**
If the paper is accepted, the camera-ready version will include (i) strengthened theoretical explanations, (ii) new experimental results further validating CGES across models and benchmarks, and (iii) improved clarity and completeness throughout the manuscript. These additions reinforce CGES as a principled, practical, and broadly effective framework for confidence-guided self-consistency.

---

### Author Response · Authors · 2025-12-04

We have incorporated all of the revisions into the updated manuscript. All modified passages, including added explanations and new experimental results, are highlighted in **blue** for ease of review.

---

### Meta-Review · Area_Chair_t3rz · 2026-01-07

**Summary:**

The reviewers raised several major concerns. First, most reviewers noted insufficient discussion of related work and the lack of comparison against many existing baselines. Second, reviewers raised concerns regarding experimental evaluation, in terms of the scale of the experiments, the number of repetitions required for robust evaluation, and missing ablations. Third, reviewer efJq raised serious concerns regarding the validity of the assumptions underlying the theoretical results, as well as unclear mathematical definitions and derivations. Additionally, reviewers raised several other clarification questions.

**Reviewer Concerns:**

The rebuttal partially addressed reviewers' concerns by providing a more detailed discussion of related work and additional experimental results. One additional baseline (ASC) was included, and the authors indicated that they plan to include another requested baseline (DSC); however, the performance of the proposed method relative to DSC remains unclear at this stage. The rebuttal also mentioned ongoing work on using subsampling to provide more robust experimental evaluations, although no corresponding results were provided.

The rebuttal also partially address reviewer efJq's concerns by clarifying definitions and derivations, and providing additional explanation of the assumptions used in theoretical analysis. The rebuttal acknowledged that some of the assumptions are idealized and used solely for the derivation of Theorem 1. However, this clarification may raise additional questions regarding the role of Theorem 1 and its relationship to Theorem 2 and the actual algorithm.

**Reviewer Scores:**

Given that the rebuttal is unlikely to have fully addressed the concerns raised by Reviewers efJq and 5UEf, the AC believes that they would likely have maintained a leaning toward rejection of the current submission. It remains unclear whether Reviewer efJq would have been inclined to revise their score upward (e.g., from 2 to 4).

---

### Decision · Program_Chairs · 2026-01-26

Reject